# Functional reconstruction of a eukaryotic-like E1/E2/(RING) E3 ubiquitylation cascade from an uncultured archaeon

Rory Hennell James[1], Eva F. Caceres[2], Alex Escasinas[3], Haya Alhasan[3], Julie A. Howard[4], Michael J. Deery[4], Thijs J.G. Ettema [2] & Nicholas P. Robinson [3]

The covalent modification of protein substrates by ubiquitin regulates a diverse range of critical biological functions. Although it has been established that ubiquitin-like modifiers evolved from prokaryotic sulphur transfer proteins it is less clear how complex eukaryotic ubiquitylation system arose and diversified from these prokaryotic antecedents. The discovery of ubiquitin, E1-like, E2-like and small-RING finger (srfp) protein components in the Aigarchaeota and the Asgard archaea superphyla has provided a substantive step toward addressing this evolutionary question. Encoded in operons, these components are likely representative of the progenitor apparatus that founded the modern eukaryotic ubiquitin modification systems. Here we report that these proteins from the archaeon *Candidatus 'Caldiarchaeum subterraneum'* operate together as a bona fide ubiquitin modification system, mediating a sequential ubiquitylation cascade reminiscent of the eukaryotic process. Our observations support the hypothesis that complex eukaryotic ubiquitylation signalling pathways have developed from compact systems originally inherited from an archaeal ancestor.

[1] Department of Biochemistry, The University of Cambridge, Cambridge, CB2 1GA, UK. [2] Department of Cell and Molecular Biology, Science for Life Laboratory, Uppsala University, Uppsala, 751 24, Sweden. [3] Division of Biomedical and Life Sciences, Faculty of Health and Medicine, Lancaster University, Lancaster, LA1 4YG, UK. [4] Department of Biochemistry and Cambridge Systems Biology Centre, Cambridge Centre for Proteomics, The University of Cambridge, Cambridge, CB2 1GA, UK. Correspondence and requests for materials should be addressed to N.P.R. (email: n.robinson2@lancaster.ac.uk)

The complex ubiquitylation systems of eukaryotes orchestrate diverse regulatory cell signalling pathways that play instrumental roles in maintaining cellular viability. It is well documented that the attachment of the ubiquitin small modifier is central to proteasomal degradation pathways, transcriptional control, DNA repair, cell cycle regulation and a plethora of other regulatory pathways[1–8]. How these ubiquitylation systems were acquired by the earliest eukaryotes and then subsequently developed into the sophisticated apparatus employed by the most

complex descendants has garnered considerable interest of late. Indeed, it has been hypothesised that the acquisition of a primitive prokaryotic ubiquitylation system may have been a prerequisite to permit the endosymbiotic event central to eukaryogenesis[9]. Furthermore, the diversification of this protein modification system has been shown to be concomitant with the increasing cellular complexity during eukaryotic evolution[10–14].

The canonical eukaryotic ubiquitylation process involves three key biochemical steps, operating in a sequential cascade, which

**Fig. 1** Operonic arrangement of the putative archaeal ubiquitylation apparatus. **a** Gene clusters of the putative ubiquitin-like protein modifier system identified in archaeal species: the ubiquitin, E1-like, E2-like and small RING finger protein (srfp) components are coloured as indicated in the key. The Cluster I and II division is described in Supplementary Fig. 1, which provides a more comprehensive analysis showing an additional Cluster III. See Methods and Supplementary Table 2 for more details on the E1-like C-terminal ubiquitin-like (UBL) domain identification. (boxed inset): Operonic arrangement of the genes encoding the components of the *C. subterraneum* ubiquitylation cascade investigated in this study. The Rpn11/JAMM domain metalloprotease homologue is juxtaposed to the operon and transcribed right to left, whereas the ubiquitylation operon is transcribed left to right

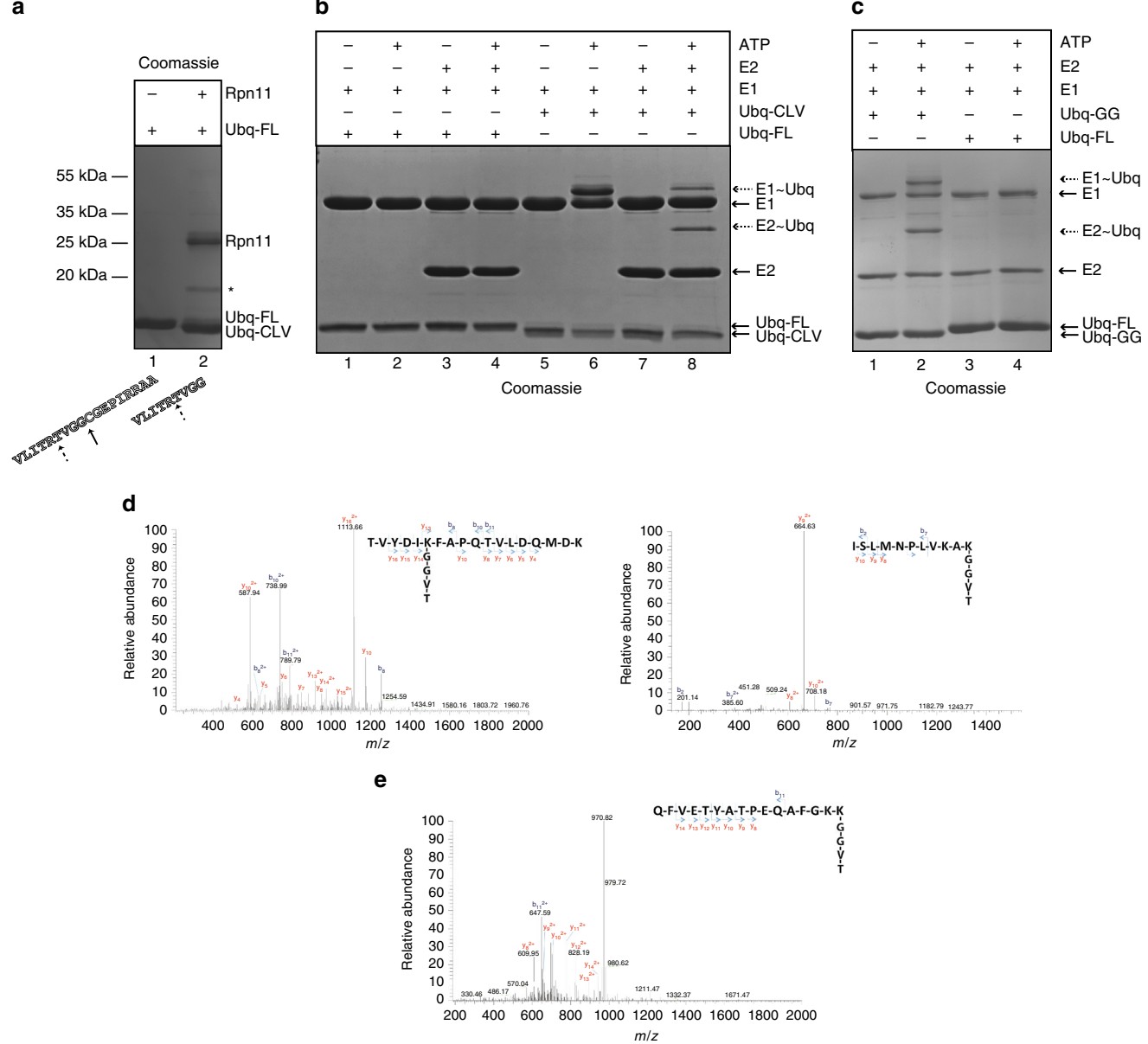

**Fig. 2** Rpn11/JAMM metalloprotease cleavage of the *C. subterraneum* ubiquitin precursor to generate a canonical C-terminal di-glycine motif that is subsequently activated by the E1-like enzyme. **a** *C. subterraneum* Rpn11 specific cleavage of the C-terminus of the pro-ubiquitin generates a mature ubiquitin species. Lane 1: pro-ubiquitin only; lane 2: pro-ubiquitin plus Rpn11. (Asterisk: C-terminal truncation of Rpn11). **b** Auto-ubiquitylation of the E1-like and E2-like components by ATP-dependent conjugation of the cleaved ubiquitin species via the exposed ubiquitin di-glycine motif. Lane 1 and 2: uncleaved, full-length, pro-ubiquitin (Ubq-FL) plus E1 in the presence and absence of ATP, respectively; lanes 3 and 4 as in lanes 1 and 2 but with the inclusion of the E2-enzyme; lanes 5–8 as in lanes 1–4 but using Rpn11-cleaved ubiquitin (Ubq-CLV) (assay at 60 °C for 15 min). **c** Ubiquitin truncated by a stop-codon after the di-glycine motif (Ubq-GG) also auto-ubiquitylates the E1-like and E2-like proteins. Reactions using Ubq-GG in the absence or presence of ATP are displayed in lanes 1 and 2, respectively (*cf* Ubq-FL shown in lanes 3 and 4). Reactions were performed as described in **b**. Products in **a**, **b** and **c** separated by SDS-PAGE and visualised by Coomassie staining. The C-terminal Rpn11-specific cleavage site in pro-ubiquitin is displayed below panel **a** (solid arrow). Trypsin digestion (dashed arrow) results in the generation of the 'TVGG' moiety detected as an isopeptide linkage in the tandem mass-spectrometry analyses. **d** MS/MS spectra of the auto-ubiquitylated E1 peptides following trypsin digestion to generate the 'TVGG' isopeptide linked moiety. m/z values of the precursor ions are shown in the top left of each panel. (2+) indicates doubly charged precursor ions. Spectra show the annotated peaks that are due to C-terminal y (coloured red) and N-terminal b (coloured blue) fragment ions. In each case, the m/z values of the precursor ions and the m/z values of the fragment ions are consistent with lysine residues modified with the 'TVGG' moiety. The amino-acid sequence of the trypsin-generated peptide, including the 'TVGG' modified lysine, is shown within in each example. **e** MS/MS spectra of the auto-ubiquitylated E2 peptide, as described in **d**

ultimately results in the covalent attachment of the small ubiquitin modifier to a target lysine on a substrate. The first protein, referred to as the E1 enzyme, activates the modifier by adenylating the C-terminal residue of the di-glycine motif that is a feature of almost all ubiquitin-like modifier proteins[15, 16]. The activated modifier is then transferred to the catalytic cysteine of the E1 enzyme, forming a covalent thioester intermediate[4]. Upon interaction with a second protein, the E2 conjugating enzyme, this activated ubiquitin moiety is then shuttled from the catalytic centre of the E1 protein to the catalytic cysteine of the E2 protein

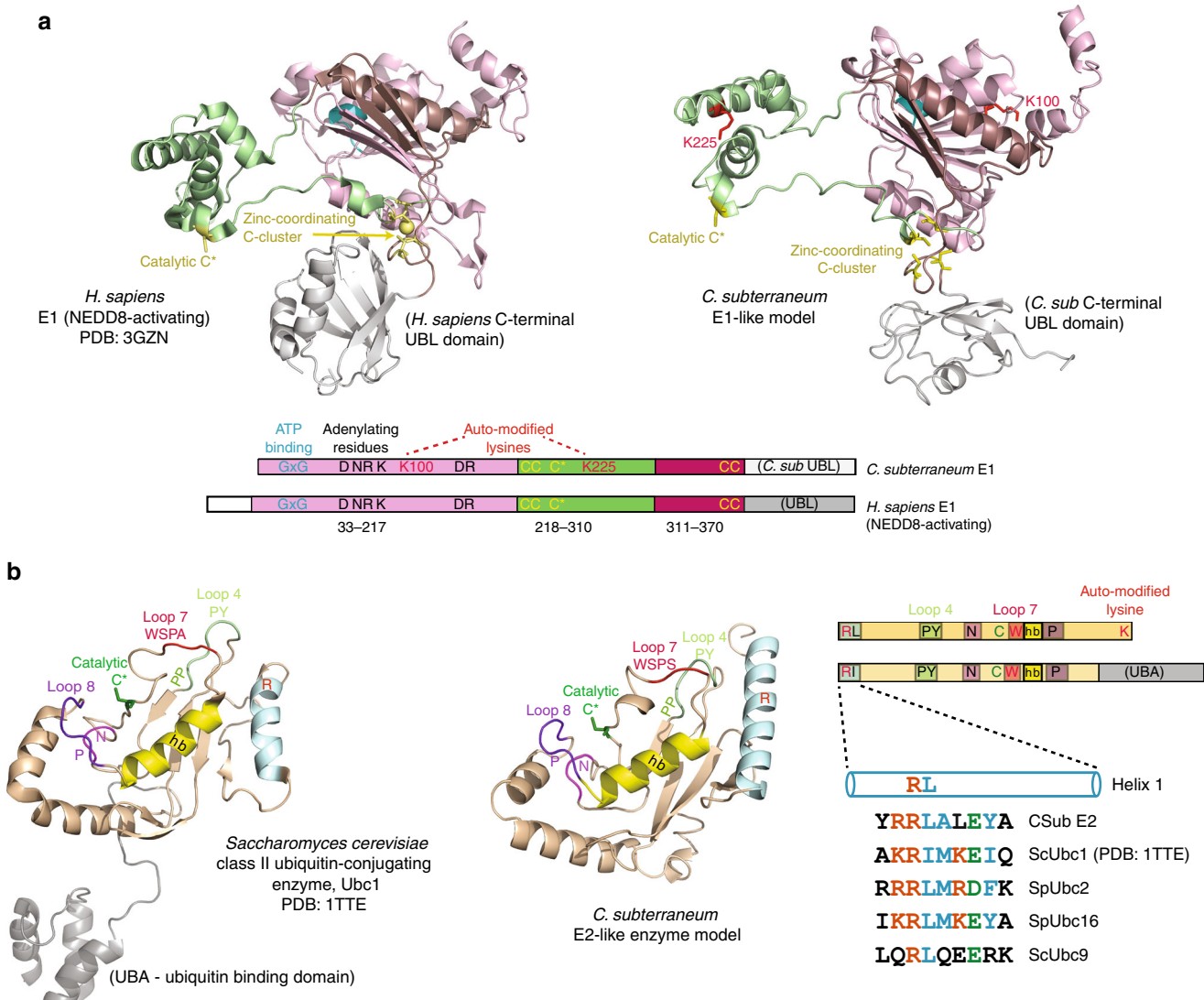

**Fig. 3** Structural features and key residues conserved between the *C. subterraneum* and eukaryotic E1-like and E2-like homologues. **a** top left; Crystal structure of the *H. sapiens* NEDD8 activating E1-enzyme (PDB: 3GZN) predicted as a structural homologue of the *C. subterraneum* E1-like enzyme by PHYRE2; top right I-TASSER model of the *C. subterraneum* E1-like enzyme. Active and inactive Rossman fold domains are coloured dark and light salmon, respectively; the active cysteine-containing domain is coloured *green* (key cysteines are shown as gold sticks); the ubiquitin-like (UBL) domains are coloured grey. A coordinated zinc-ion is represented by a gold sphere. The two specific auto-ubiquitylated lysine residues in the *C. subterraneum* structure are shown as red sticks. The glycine rich ATP-binding loop is coloured blue. bottom, schematic representation of the *Homo sapiens* and *C. subterraneum* E1-like enzymes with colours as described in **a**. Yellow 'C's represent the zinc-binding and catalytic cysteine residues. Blue 'GxG' lettering denotes the glycine-rich ATP-binding loop. Conserved 'D', 'NR', 'K' and 'DR' residues are critical for ubiquitin adenylation in eukaryotic E1-enzymes[38] are indicated (see Supplementary Notes 2 and 3 for further details). **b** left; crystal structure of the *S. cerevisiae* Ubc1 E2-enzyme (PDB: 1TTE), a PHYRE2 predicted homologue of the *C. subterraneum* E1-like enzyme. Middle: I-TASSER model of the *C. subterraneum* E2-like enzyme. The N-terminal E1-interacting helix is coloured light blue ('R' indicates an invariant arginine), the hydrophobic cross-over helix (labelled 'hb') is coloured yellow; the catalytic cysteine shown as green sticks (labelled 'C'). The HPN loop is labelled 'N' and coloured magenta. Loop 4 is coloured light green and labelled 'PS', while loop 7 is coloured red and labelled 'WSPA/S'. Loop 8 is coloured purple (labelled 'P') and the *S. cerevisiae* UBA domain is shaded grey. Structural figures generated using PyMOL. right; schematic representation of the *S. cerevisiae* and *C. subterraneum* E1-like enzymes with the same colours and labels. The consensus sequence of the conserved E1 (or E3) interacting motifs for the *C. subterraneum* E2-like enzyme, and yeast E2 homologues (*S. cerevisiae* Ubc1 and Ubc9 and *S. pombe* Ubc2 and Ubc16) are illustrated. Structural figures were prepared using PyMOL

via a trans-thioesterification event[4, 17, 18]. It has been revealed that dramatic conformational changes within the E1 enzyme are coincident with these two sequential thioesterification events during which a second ubiquitin moiety is bound to the activating-enzyme, facilitating the transfer of the first ubiquitin to the E2-enzyme[17, 18]. The next stage in the ubiquitylation cascade is mediated by the E3 ligase[19–21]. This protein stimulates the activity of the E2 conjugating enzyme and catalyses the amino-

lysis based transfer of the ubiquitin from the E2 enzyme onto a specific lysine residue on the target protein[18]. The cascade ultimately results in an isopeptide bond formed between the primary amino group on a lysine residue on the substrate and the carboxyl group of the ubiquitin moiety[2]. While hundreds of E3 enzymes have been identified to date, these enzymes can be divided into three broad classes[20]. RING-E3 enzymes mediate direct transfer of ubiquitin from the E2 enzyme to the target, and simultaneously

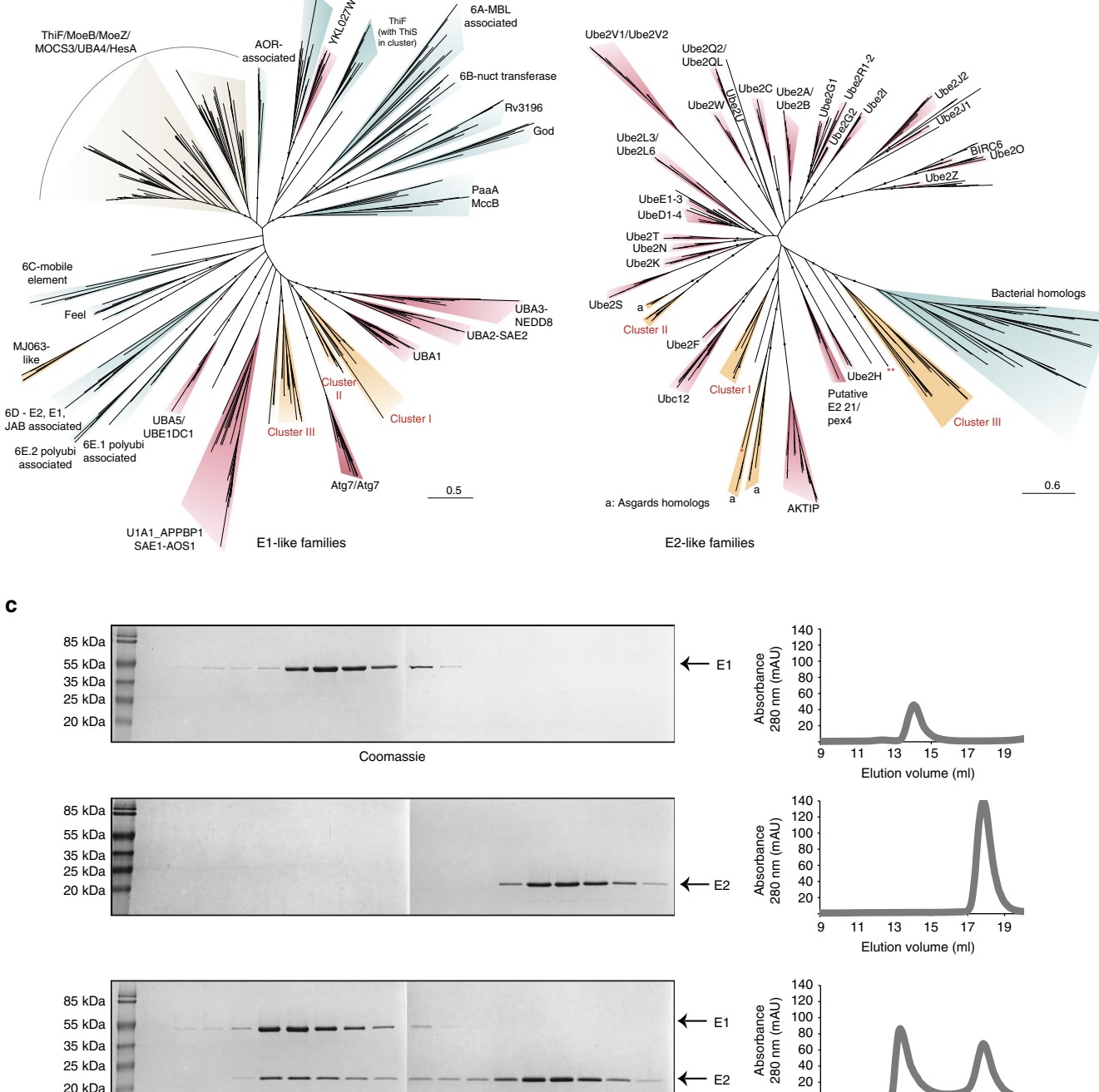

**Fig. 4** Phylogenetic comparisons between the archaeal and eukaryotic E1 and E2 homologues and demonstration of the *C. subterraneum* E1-E2 interaction and complex formation by size exclusion chromatography. **a–b** Phylogenetic analysis of the E1-like **a** and E2-like **b** homologues. Unrooted maximum likelihood trees inferred with iqtree and the LG + R8 **a** and LG + R6 **b** models. Ultrafast-bootstrap support values for the deepest nodes are indicated by empty (UFBoot ≥ 95) and filled circles (UFBoot ≥ 99). Major clades of the E1 and E2 phylogenies are named according to the nomenclature used by Burroughs et al.[38] and Stewart et al.[18], respectively. Clades are colour-coded according their taxonomic affiliation: eukaryotes (red), bacteria (blue), archaea (orange) and mixed (grey). Asterisks denote E2-like homologues described further in Supplementary Fig. 1. Abbreviations: a, Asgard archaea homologues. **c** Physical interaction between the *C. subterraneum* E1 and E2 components demonstrated by size exclusion chromatography. left: *C. subterraneum* E1-like enzyme only (top), E2-like protein only (middle), or pre-incubated E1-like and E2-like proteins (bottom) were separated on a superdex S200 HR 10/300 size exclusion chromatography column. The relative elution volumes of the size standards bovine serum albumin (BSA) (66 kDa) and carbonic anhydrase (29 kDa) are also indicated (in grey). Eluted fractions were resolved by SDS-PAGE and visualised by Coomassie stain. Right: chromatography UV traces (at 280 nm) for the respective elution profiles

bind the E2 enzyme and the substrate[22]. By contrast, the HECT (homology to E6AP C terminus)[23] and RBR (RING-between-RING)[24] classes of E3 enzymes catalyse an additional trans-thioesterification event during which the ubiquitin is passed to an additional catalytic cysteine residue within the E3 enzyme itself before the final transfer to the target lysine on the substrate[20]. The selective pairing between the E2 conjugating enzymes and their multiple cognate E3 ligases confers the

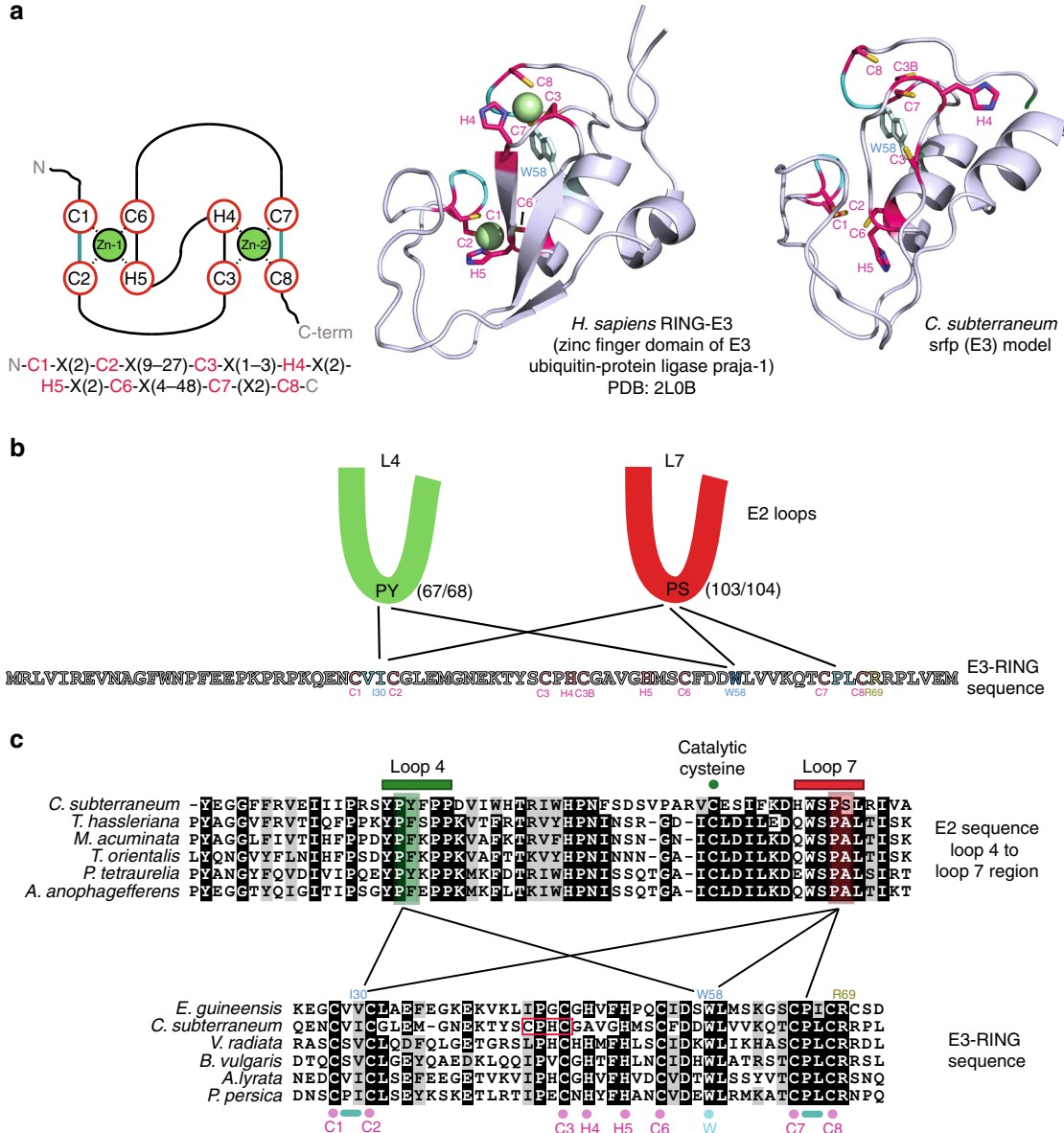

**Fig. 5** Structural prediction of the canonical RING-domain crossbrace arrangement of the *C. subterraneum* srfp (E3) protein. **a** (left top) schematic representation of the RING-domain cross-brace structure with the zinc-coordinating residues circled in red; (left bottom) consensus sequence of the RING-domain (RING-H2 family) with zinc-binding residues highlighted in red ('X' any residue); (middle top): RING-domain of the *Homo sapiens* RING-E3 ubiquitin ligase praja-1 (PDB: 2L0B); (right top): I-TASSER model of the *C. subterraneum* srfp (E3-like) homologue. Zinc-binding residues are coloured pink (with carbon and sulphur atoms in blue and yellow, respectively) and a conserved tryptophan coloured blue. Green spheres denote $Zn^{2+}$ ions. Loops that interact with conserved residues in the E2-enzymes are coloured blue. **b** Schematic predicting contacts between residues in the *C. subterraneum* RING domain and the E2-like enzyme loops 4 and 7 based upon conservation of residues in eukaryotic components[21, 22, 41, 44]. Zinc-coordinating residues are coloured red; the conserved tryptophan and the hydrophobic residues presented on loops in the RING domain are coloured blue. Key conserved hydrophobic residues on the E2-like enzyme (PY or PS on loops 4 or 7, respectively) are indicated. **c** Multiple sequence alignments of the *C. subterraneum* E2-like and E3-like protein amino-acid sequences with the closest eukaryotic homologues also indicating the specific E2/E3 interactions shown in **b**. E2-ubiquitin-conjugating homologues: (*Paramecium tetraurelia* [Alveolata ciliate], *Musa acuminata* [banana], *Aureococcus anophagefferens* [alga], *Tarenaya hassleriana* [spider flower], *Theileria orientalis* [Apicomplexan parasite]). E3-RING homologues: (*Elaeis guineensis* [oil-palm], *Vigna radiata* [mung bean], *Beta vulgaris* [sugar beet], *Arabidopsis lyrata* [rockcress], *Prunus persica* [peach]). The conserved motifs on loops 4 and 7 of the E2-like enzymes are represented by green and red bars, respectively. The catalytic cysteine is denoted by a green circle. The E3-like zinc-binding cysteines and histidines are highlighted by purple circles. Note that the position of the *C. subterraneum* C3 and H4 are out of alignment by 2 residues (red box). The conserved tryptophan (blue circle) and hydrophobic interaction interfaces between cysteines C1 and C2 and C7 and C8 (blue bars) are also highlighted. The position of residues I30, W58 and R69 are also indicated. Structural figures were prepared using PyMOL

specificity necessary for the varied biological pathways regulated by ubiquitylation[20, 22].

The identification of an operon encoding a eukaryotic-like ubiquitylation system in the genome of the uncultured archaeon Candidatus 'Caldiarchaeum subterraneum' (hereafter *C. subterraneum*) has led to the proposal that a streamlined ubiquitylation modification system may have been inherited during eukaryogenesis from an archaeal progenitor[25]. More recent bioinformatics studies have provided further data that is supportive of this assumption[11, 25–28]. While primitive archaeal ubiquitin-like modifications have been reported previously, these pathways are believed to be dependent on just an E1 enzyme[29, 30], reminiscent of the mechanisms utilised by prokaryotic sulphur transfer pathways from which they evolved[31, 32]. In contrast, a complete prokaryotic ubiquitylation cascade utilising all three enzymatic steps has not been demonstrated experimentally to date. In this study we synthetically reconstitute and biochemically characterize the four components encoded within the ubiquitylation operon and also the associated Rpn11 metalloprotease enzyme. We reveal that the *C. subterraneum* genome does indeed encode a fully functional and minimal ubiquitylation system. We compare our findings to the previously described eukaryotic ubiquitylation cascades and discuss our results in the context of the evolution of eukaryotic ubiquitylation systems.

## Results

### Cleavage of the *C. subterraneum* pro-ubiquitin by Rpn11. 
To demonstrate functional activity of the Candidatus 'Caldiarchaeum subterraneum' (hereafter *C. subterraneum*) ubiquitylation operon, we synthesised the genes encoding ubiquitin, E1-like, E2-like and srfp (E3-like) components and expressed and purified these proteins. The *C. subterraneum* ubiquitin homologue is encoded in the reconstructed composite genome[25] as a pro-ubiquitin with a nine amino-acid peptide extending beyond the conserved C-terminal di-glycine motif. Similarly, in eukaryotes, ubiquitin and many other ubiquitin-like modifiers such as SUMO (small ubiquitin-like modifier) and NEDD8 (neural precursor cell-expressed, developmentally downregulated)[33] are commonly expressed as fusion proteins or pro-ubiquitin-like precursors, and the generation of the mature ubiquitin moiety requires the action of a dedicated protease[34]. It therefore followed that the *C. subterraneum* C-terminal pro-peptide must be removed to generate the mature active ubiquitin. We predicted that the Rpn11 metalloprotease homologue, encoded close to the ubiquitylation operon in *C. subterraneum* and a number of other archaeal species (Fig. 1 and Supplementary Fig. 1), would function to generate the mature ubiquitin species. Combination of the pro-ubiquitin and the Rpn11-like homologues in a proteolytic processing assay revealed that the metalloprotease enzyme was competent to cleave the pro-ubiquitin (Fig. 2a). Tandem mass spectrometry analysis of the cleaved product confirmed that the nine-amino acid pro-peptide was removed, exposing the di-glycine motif of the modifier (see Supplementary Note 1). Furthermore, we revealed that while the full-length pro-ubiquitin was refractory to activation by the E1 enzyme as predicted, Rpn11-cleaved ubiquitin, or ubiquitin C-terminally truncated immediately after the di-glycine motif by the introduction of a stop-codon, were activated and the reaction resulted in auto-ubiquitylation of the E1 protein itself (Fig. 2b, c). Tandem mass spectrometry analysis of the modified product confirmed that the ubiquitin moiety was covalently attached via an isopeptide bond between the terminal glycine and specific lysine residues on the E1 enzyme (Fig. 2d). Ubiquitin-like auto-modifications of archaeal E1-like enzymes have been observed previously[29, 30] (see Supplementary Note 2 for further discussion). Addition of the

E2-like conjugating enzyme to the E1-dependent ubiquitin activation reaction resulted in the generation of another distinct ubiquitylated product (Fig. 2b, c). Tandem mass spectrometry revealed this product to be a specific auto-mono-ubiquitylation on a C-terminal lysine on the E2 component itself (Fig. 2e and see Supplementary Note 3 for further discussion).

**Structural conservation of the archaeal / eukaryotic enzymes**. Generation of structural models for the *C. subterraneum* E1-like and E2-like enzymes, using the I-TASSER server[35–37] (Supplementary Table 1), revealed conservation of the major structural features and key catalytic residues that are characteristic of the E1 ubiquitin-activating and the E2 ubiquitin-conjugating (UBC) domains (Fig. 3a, b, and Supplementary Figs 2 and 3, respectively)[17, 18, 38–42]. These conserved structural features shared between the archaeal and eukaryotic E1-like and E2-like enzymes are central to the eukaryotic ubiquitylation cascade. The I-TASSER model of the *C. subterraneum* E2 enzyme revealed that the first N-terminal alpha-helix harboured a motif that matched the consensus sequence observed in the equivalent helix of the eukaryotic E2 homologues (Fig. 3b). This region is known to mediate interaction with the ubiquitin-binding domain of the eukaryotic E1-enzyme, and also forms the canonical binding site with E3 enzymes[18, 22, 43]. The model also revealed conservation of the essential loops 4 and 7 (including the PxxPP and D/ExWSP motifs; Fig. 3b), which have been proposed to play essential roles in the specific interactions with the cognate E3 ligases in eukaryotic systems[41, 44] as discussed further in the Supplementary Note 3. Furthermore, phylogenetic analyses of the E1-like and E2-like proteins confirmed the close evolutionary relationship between the *C. subterraneum* and eukaryotic homologues (Fig. 4a, b, respectively and additional discussion in the Supplementary Note 4). In addition to the functional biochemical activity identified in the auto-ubiquitylation assays, a stable physical interaction between the *C. subterraneum* E1 and E2 components was also observed by analytical size exclusion chromatography (Fig. 4c).

The overall similarity in structural arrangement of the eukaryotic and archaeal E2 enzymes extended to the conservation of key amino-acid residues that mediate the interaction with E3 ligases[21, 22, 41] (Fig. 3b and Supplementary Figs 3 and 4). We therefore also compared the *C. subterraneum* E3-like srfp protein with known structural homologues to search for the reciprocal conserved regions on the archaeal ligase that might mediate these interactions. It has been established that eukaryotic RING-domain proteins fold with a cross-brace arrangement by coordinating $Zn^{2+}$ ions, forming an interface for E2 binding[21, 22, 45]. The PHYRE2 protein fold recognition server[46] identified the zinc finger domain of the *Homo sapiens* E3 ubiquitin-protein ligase Praja-1 (PDB code: 2L0B) as a top structural homologue (Fig. 5a and Supplementary Table 1). Furthermore, the I-TASSER structural prediction software generated a model for the *C. subterraneum* homologue, which revealed that the key residues and surfaces involved in the interaction with the E2 conjugating enzyme were also conserved (Fig. 5a–c and Supplementary Figs 3 and 4). The conserved zinc-coordinating cysteine and histidine residues indicated that the *C. subterraneum* srfp protein belongs to the RING-H2 class of proteins, defined by a C3H2C3 sequence of zinc-binding residues (Fig. 5a)[21]. However, it should be noted that the position of the *C. subterraneum* C3 and H4 residues are out of alignment by two residues and an additional cysteine residue (denoted C3B) may be involved in the coordination of the second zinc ion, as predicted in the I-TASSER model (Fig. 5a–c). As suggested in previous studies in eukaryotic systems[45, 47–49], it seems plausible that the

conserved residues P67/Y68 and P103/S104 on the *C. subterraneum* E2-enzyme on loops L4 and L7, respectively, are critical for the interaction with the E3 components (Figs. 5b, c and Supplementary Figs 3 and 4). These residues associate either with conserved hydrophobic residues that are located on two

loops on the E3 protein, or with an invariant tryptophan on the intervening α-helix, which contributes to the E2 interaction surface (Fig. 5a–c)[22]. In eukaryotic systems it had previously been demonstrated that the docking of the E3 RING domain onto the cognate E2-conjugating enzyme is able to dramatically stimulate

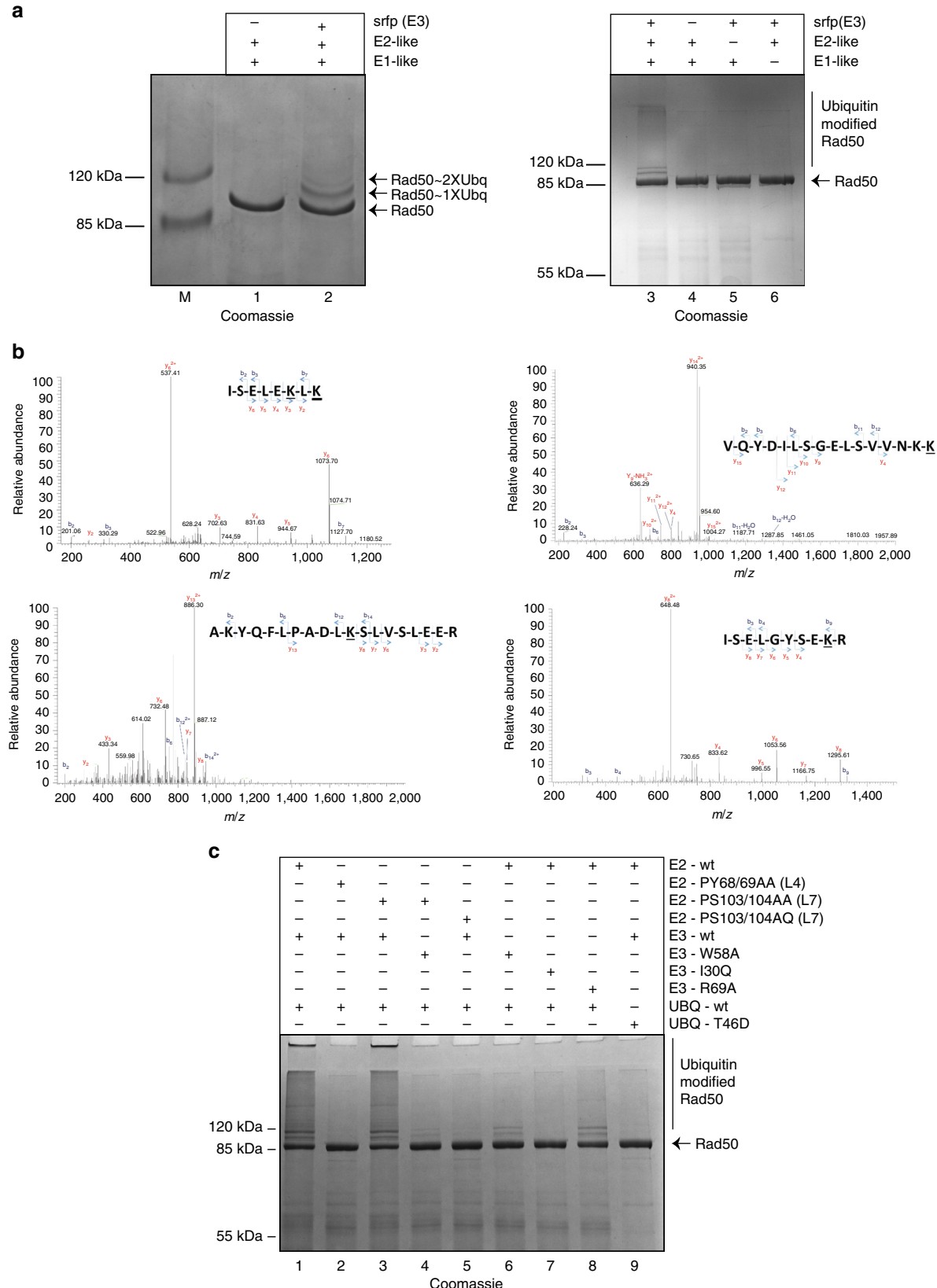

the activity of the E2-enzyme by restricting movement of the ubiquitin-E2 thioester intermediate, inducing a conformationally constrained or 'closed' state in which it is positioned ideally for the catalytic transfer to a lysine on the substrate[50, 51]. The conservation of the key residues and structural features on the reciprocal binding surfaces of the *C. subterraneum* E2 and srfp-E3 proteins predicted in the I-TASSER models suggested that the archaeal ubiquitylation system also operates by an equivalent mechanism. We sought to verify these bioinformatics predictions experimentally by attempting to biochemically reconstitute the E1/E2/srfp(E3) ubiquitylation cascade and then probe the system using site-directed mutations of the key residues.

**Reconstitution of the E1/E2/(RING)E3 ubiquitylation cascade.**
As, to date, *C. subterraneum* is an uncultured organism with its composite genome reconstructed from metagenomic data[25], it is currently not possible to determine the physiological substrates acted upon by the ubiquitylation cascade. In the absence of a known candidate, we decided to use the thermophilic Mre11/Rad50 complex from the thermophilic crenarchaeote *Sulfolobus acidocaldarius* as a substrate. We have previously shown that the Rad50 component of this bi-partite DNA repair complex was extensively modified by the Urm1/SAMP protein following *in vivo* overexpression of this ubiquitin-like modifier in *S. acidocaldarius* cells[30]. Inclusion of the Mre11/Rad50 complex in a reconstituted ubiquitylation assay with the *C. subterraneum* ubiquitin, E1-like, E2-like and srfp (E3-like) components resulted in the covalent modification of the *S. acidocaldarius* Rad50 protein (Fig. 6a). Crucially, these modifications were dependent on the addition of the srfp E3-like homologue. Indeed, if any one of the E1-like/E2-like/srfp (E3-like) enzymes was removed from the reaction, we could no longer observe the formation of the conjugates (Fig. 6a), which were detected by both Coomassie staining and tandem mass spectrometry analyses (Fig. 6a, b). It was therefore evident that the conjugates were generated via a canonical eukaryotic-like ubiquitylation cascade. The ubiquitylation assays identified at least two major products above 100 kDa consistent with mono and di-ubiquitylations, respectively. Furthermore, smeared products visible in the region above the two discrete modified bands were suggestive of ubiquitin chain formation (Figs. 6a, c). Tandem mass-spectrometry analysis of the modified region of the gel above the native Rad50 band revealed conjugations of ubiquitin on four specific lysine residues within the coiled-coil regions on the Rad50 substrate (K256, K600, K618, K662) (Fig. 6b).

In order to investigate the role of the predicted key residues involved in the ubiquitylation cascades, site-directed mutants of the E2-like, srfp (E3-like) and ubiquitin proteins were generated

and used in the functional assay (Fig. 6c). The E2-PY67/68AA (loop 4) and E2-PS103/104AA (loop7) mutations, along with the complementary srfp (E3-like) mutations W58A and I30Q were examined in order to explore the predicted E2/E3 protein-protein interface (Fig. 5b, c). Substituting the E2-PY67/68AA and srfp (E3-like)-I30Q mutant components in the place of the native proteins resulted in an abrogation of the ubiquitylation reaction (Fig. 6c; lanes 2 and 7, respectively), while the srfp (E3like)-W58A mutation also significantly impaired the ubiquitylation reaction. Although mono- and di-ubiquitylations were detected when the srfp (E3-like)-W58A mutant was tested (Fig. 6c; lane 6) the long smear suggestive of ubiquitin chain elongation was no longer detectable. In contrast to the E2-PY67/68AA (E2-loop 4) mutation, the E2-PS103/104AA (E2-loop 7) mutation did not appear to affect the efficiency of the ubiquitylation reaction when compared to the wild-type control (Fig. 6c; lane 3). However, combination of both the E2-PS103/104AA (E2-loop 7) and srfp (E3like)-W58A mutations resulted in an inhibition of the ubiquitylation reaction that was more pronounced than the effect observed with W58A mutation alone (Fig. 6; lane 4, compared with lane 6). Since, the S104A substitution in the E2-PS103/104AA (E2-loop 7) loop mutant was a conserved change, and an alanine was often encoded in the equivalent position in eukaryotic homologues (Fig. 5c), an E2-PS103/104AQ (E2-loop 7) mutant in which the serine was instead substituted for the bulky polar residue glutamine was also investigated. This mutation resulted in considerable impairment of the ubiquitylation reaction (Fig. 6; lane 5). These results were therefore suggestive that the conserved residues on both loops 4 and 7 of the E2-like enzyme, and the corresponding I30 and W58 residues on the srfp (E3like), do indeed play a critical role in the interaction between these two components and regulate the subsequent ubiquitylation process, as seen for the eukaryotic apparatus[21].

An additional mutation in the srfp (E3-like) protein, R69A, was also found to impair the ubiquitylation cascade (Fig. 6c; lane 8). This residue is conserved (as either a lysine or arginine) in all E3-RING proteins and plays a critical role in the ubiquitylation process by regulating formation of the conformationally constrained 'closed' state of the E2-ubiquitin intermediate required for efficient chain elongation[51]. In the light of the R69A result it seems likely that this mechanism also operates in the conserved archaeal systems. Indeed, in further support of this assumption we demonstrated that mutation of the hydrophobic patch on the surface of ubiquitin, a region that is essential for the formation of the 'closed' E2-ubiquitin intermediate state[51], also resulted in a dramatic impairment of the ubiquitylation reaction (Fig. 6c, lane 9).

**Fig. 6** Requirement of the E3 component for the in vitro ubiquitylation of a Rad50 substrate from *Sulfolobus solfataricus* and investigation of the key residues predicted by the structural models to be required for the functional enzymatic cascade. **a** The *C. subterraneum* srfp (E3-like) protein is required for the in vitro ubiquitylation of a *S. solfataricus* Rad50 substrate. (left) Lanes 1 and 2: ubiquitylation reaction containing Rad50 substrate and the ubiquitin, E1-like, E2-like proteins in the absence or presence of the srfp (E3-like) protein, respectively. ubiquitylation assay and srfp (E3-like) proteins required for the ubiquitylation cascade. (right) exclusion of any individual component from the reaction prevents ubiquitylation of the Rad50 component. The reaction in lanes 4 lacks the srfp (E3-like) component, while the reactions in lane 5 and 6 lack the E2-like and E3-like enzymes, respectively. **b** MS/MS spectra of the ubiquitylated *S. solfataricus* Rad50 peptides identified in the modified bands. (2+) indicates doubly charged precursor ions. Spectra show the annotated peaks that are due to C-terminal y (coloured red) and N-terminal b (coloured blue) fragment ions. In each case, the m/z values of the precursor ions and the *m/z* values of the fragment ions are consistent with ubiquitylations. The lysine modified with the 'TVGG' isopeptide-linked moiety, generated following trypsin digestion, is underlined. **c** Examining the importance of predicted key residues in the ubiquitin, E1-like and E2-like proteins using the functional in vitro ubiquitylation assay of the *S. solfataricus* Rad50 substrate. All reactions included the ubiquitin, E1-like, E2-like and srfp (E3)-like components. Mutant proteins were substituted for the native (wild-type) proteins as indicated. Lane 1, native protein (wild-type) proteins only (control); lane 2, E2-PY67/68AA (loop 4) mutant; lane 3, E2-PS103/104AA (loop7) mutant; lane 4, combination of E2-PS103/104AA (loop7) and E3-W58A mutants; lane 5, E2-PS103/104AQ mutant; lane 6, E3-W58A mutant; lane 7, E3-I30Q mutant; lane 8, E3-R69A mutant; lane 9, ubiquitin-T46D mutant. All assays were performed at 60 °C for 60 min as described in the methods section. Products were separated by SDS-PAGE and visualised by Coomassie staining

**Ubiquitin chain linkages arising from the enzymatic cascade.** In addition to the modifications identified on the Rad50 substrate, three ubiquitylated lysine residues (K4, K53, K68) were also observed by tandem mass-spectrometry on the ubiquitin moiety itself (Fig. 7a, b), consistent with the possibility of ubiquitin chain formation. Comparison of the crystal structure of human ubiquitin (PDB code: 1UBQ) with the available *C. subterraneum* NMR solution structure (PDB code: 2MQJ) revealed that the

modification on residue K53 was located at a position that appeared to be positioned closely to the highly-characterised K48 ubiquitin modification site in eukaryotic homologues (Fig. 7c, d), which has well-established roles as a protein degradation signal[3]. In addition, the modification on residue K68 of the *C. subterraneum* ubiquitin was located on strand β5 (Fig. 7d, f) while the eukaryotic K63 modification, which functions in membrane protein trafficking, immune response and DNA repair (Fig. 7c, d)

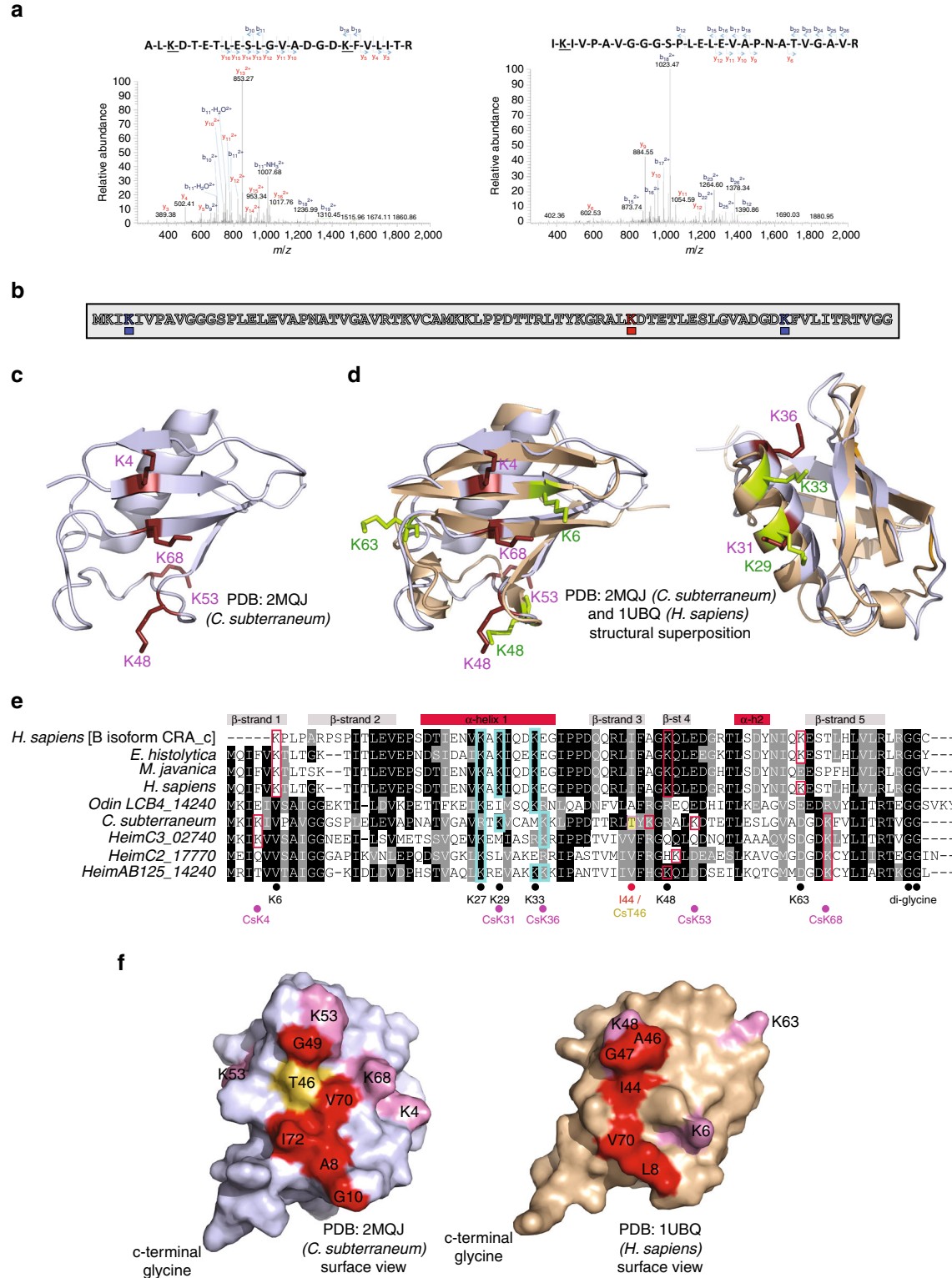

[3] is juxtaposed to strand β5 and located on the same face of the ubiquitin moiety (Fig. 7d, f). The third modification site (K4) occupies a similar position on the surface of the molecule on strand β1 of the structure that abuts the K68 residue, which lies on the adjoining strand β5 of the β-grasp fold (Fig. 7c, d). Interestingly, the eukaryotic K6 residue, which is modified to form atypical chains[52] is also positioned similarly in both the archaeal and eukaryotic homologues, located on the β1-strand of the β-sheet in both structures. We also observed that residues K31 and K36 on the *C. subterraneum* ubiquitin structure occupied positions on the first α—helix that appeared structurally equivalent to the residues K29 and K33 of human ubiquitin that are involved in forming other atypical ubiquitin chain linkages (Fig. 7d, e)[52].

The three distinct modification sites identified by the tandem mass-spectrometry analyses on the *C. subterraneum* ubiquitin moiety raise the intriguing possibility that different ubiquitin chain linkages could be formed by some archaeal ubiquitylation systems. Indeed, it is tempting to speculate that early eukaryotic cells arising from archaeal organisms possessing similar E1/E2/E3 ubiquitylation cascades may have diversified their ubiquitin chain repertoire to permit the regulation of diverse biological pathways. Subsequent expansion, especially of the E3 components, of this early eukaryotic ubiquitylation toolkit would lead to the evolution of the extremely complex chain-linkages observed in modern eukaryotic cells, which control a plethora of biological functions[3, 52] (discussed further in Supplementary Notes 5, 6).

Further comparison of the human and archaeal ubiquitin structures revealed similar exposed hydrophobic patches on both proteins (Fig. 7e, f). In eukaryotic systems this region forms a critical interface with the E2 enzyme during the 'closed' state required for efficient ubiquitin transfer[51]. The hydrophobic patch in human ubiquitin is centred on I44 and includes residues L8 and V70 (Fig. 7e, f). Similarly, in most archaeal homologues the I44 residue is conserved as a hydrophobic residue (most commonly an isoleucine or valine), although it is notable that the equivalent residue in the *C. subterraneum* ubiquitin is a threonine (T46), which influences the hydrophobic character of the exposed patch, composed of the residues A8, G10, V70, I72 and G49 (Fig. 7e, f). Nevertheless, as described above, mutation of this residue at the centre of this patch to a hydrophilic aspartic acid residue results in the inactivation of the *C. subterraneum* ubiquitylation reaction (Fig. 6c; lane 9).

**Rpn11 deconjugation of *C. subterraneum* ubiquitin linkages.** Having revealed that the *C. subterraneum* E1/E2/E3 cascade does appear to operate in a manner reminiscent of the eukaryotic ubiquitylation systems, potentially forming covalently linked ubiquitin chains extending from substrates, we next examined if the Rpn11/JAMM metalloprotease homologue, encoded in the genome close to the ubiquitin operon, could deconjugate these isopeptide linkages. Following retrieval by Ni-NTA IMAC from the reconstituted *in vitro* ubiquitylation reactions (see Methods for details), we determined that the modified Rad50 products could be deubiquitylated upon addition of the Rpn11 homologue (Fig. 8a). In addition, we revealed that a fusion protein consisting of a single ubiquitin moiety attached to the N-terminus of a thermally stable superfolder (sf)GFP protein could also be deconjugated by the metalloprotease (Fig. 8b and Supplementary Note 7).

## Discussion

In this study we have demonstrated that the compact ubiquitylation operon encoded in the genomes of archaea belonging to the *Aigarchaeota* [25], and also reported subsequently across several archaeal species including those of the Asgard superphylum[26, 28], operates to form a *bona fide* ubiquitylation cascade reminiscent of that in eukaryotes. We have revealed that the small RING-finger protein acts as an E3 ligase to stimulate the activity of the E2 conjugating enzyme and facilitates interaction with the substrate. This E1/E2/RING-E3 mechanism appears essentially indistinguishable from the known eukaryotic pathways previously reported and seemingly involves the same key conserved amino acid residues and structural features that are observed in both the archaeal and eukaryotic homologues[17, 21, 22, 40, 41, 43, 44]. The close phylogenetic roots of the archaeal ubiquitylation systems to eukaryotic counterparts, combined with the inherent thermo-stability and biochemical robustness of this reconstituted protein modification apparatus makes this *C. subterraneum* E1/E2/E3 cascade an attractive model for future structural and biochemical studies into the workings and intricacies of ubiquitylation processes. Furthermore, the putative alternative ubiquitin chain linkages and deubiquitlyation reactions observed during this study merit further investigation to advance our understanding of how complex ubiquitin-chain signalling evolved.

It is noteworthy that the broad substrate specificity displayed by the *C. subterraneum* Rpn11/JAMM metalloprotease homologue is also observed for other archaeal JAMM homologues[53], as discussed further in Supplementary Note 7. However, previous phylogenetic classification of this *C. subterraneum* metalloprotease homologue places the *C. subterraneum* Rpn11 homologue closely with the eukaryotic JAMM isopeptidase and DUB

**Fig. 7** Identification of ubiquitylated lysines on the *C. subterraneum* modifier protein and structural comparisons of the *C. subterraneum* and *H. sapiens* ubiquitins. **a** MS/MS spectra of ubiquitylated *C. subterraneum* ubiquitin peptides identified in the following the ubiquitylation of *S. solfataricus* Rad50. (2+) indicates doubly charged precursor ions. Spectra show the annotated peaks that are due to C-terminal y (coloured red) and N-terminal b (coloured blue) fragment ions. In each case, the m/z values of the precursor ions and the m/z values of the fragment ions are consistent with the underlined lysine residues modified with the 'TVGG' moiety generated following trypsin digestion. **b** Amino acid sequence of *C. subterraneum* ubiquitin indicating the three ubiquitylated lysine residues identified by the MS/MS data. **c** Solution (NMR) structure of the *C. subterraneum* ubiquitin protein (PDB: 2QMJ) with the modified lysine residues shown as sticks (in purple). **d** Structural superposition of the crystal structure of *H. sapiens* ubiquitin in wheat (PDB: 1UBQ) and the NMR structure of the *C. subterraneum* ubiquitin protein in blue (PDB: 2QMJ). (left) *C. subterraneum* modified lysines K4, K53, K68 (and an unmodified but conserved K48 residue) shown in purple, with *H. sapiens* conserved lysine residues K6, K48 and K63 displayed in green. (right) View looking onto α-helix 1 showing the *C. subterraneum* residues K29 and K33 (purple), which occupy positions on the α–helix structurally reminiscent of the K31 and K36 residues observed in the human ubiquitin molecule (green). **e** Multiple sequence alignments of the *C. subterraneum* ubiquitin protein amino-acid sequences with eukaryotic and archaeal homologues: *Homo sapiens* [B isoform CRA_c], *Entamoeba histolytica*, *Meloidogyne javanica*, *Homo sapiens*, *Odinarcheota* LCB4_14240, *Caldiarchaeum subterraneum*, *Heimdallarcheota* C3_02740, *Heimdallarcheota* C2_17770, *Heimdallarcheota* AB125_14240. Secondary structural features are indicated above the alignments. Black circles indicate conserved lysines in the eukaryotic proteins, while purple circles denote the *C. subterraneum* lysines displayed in (7c, 7d and 7e). The I44 and T46 residues are also highlighted. **f** Surface views of the *C. subterraneum* and *H. sapiens* ubiquitins, showing the similar exposed hydrophobic patches (red), centred on the human I44 residue and equivalent T46 residue (yellow) in *C. subterraneum*. Surface exposed lysines residues are displayed in purple

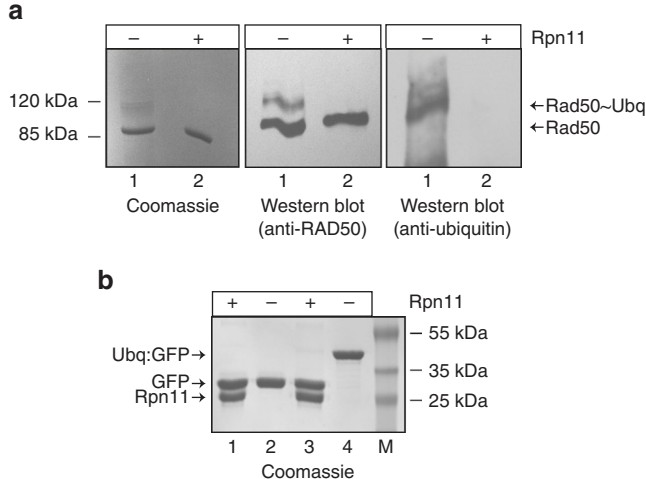

**Fig. 8** The Rpn11 JAMM metalloprotease acts as a deubiquitinase (DUB) to deconjugate isopeptide-linked ubiquitin moieties. **a** (left) Coomassie stained gel of the ubiquitylated *S. solfataricus* Rad50 product incubated either with (lane 1) or without (lane 2) Rpn11 treatment. Middle and right; western blot of duplicate gels, probed with an anti-Rad50, or anti-ubiquitin antibodies, respectively. **b** The Rpn11 homologue also cleaves ubiquitin moieties linked to the N-terminus of a substrate. Superfolder (sf)GFP or a 1xUbiquitin fusion to the N-terminus of sfGFP (1xUbq:sfGFP) incubated in the presence or absence of Rpn11. Lane 1: (sf)GFP plus Rpn11. Lane 2: sfGFP alone. Lane 3: 1xUbq:GFP following incubation with the Rpn11 protein. Lane 4: 1xUbq:GFP protein alone. Products were separated by SDS-PAGE and visualised by Coomassie staining

domains, rather than grouping with the archaeal group 1 JAMM proteases[25, 53]. Considering this close evolutionary relationship of the *C. subterraneum* deubiquitylase to eukaryotic Rpn11 (of the 19 S proteasomal lid) and Csn5 (of the COP9 signalosome) homologues, it is interesting that the archaeal homologue appears to display activity in the absence of any interacting partner protein or complex. By contrast, the aforementioned eukaryotic counterparts require incorporation within a larger multi-subunit complex (as part of the regulatory 19 S lid of the proteasome or the COP9 signalosome, respectively) in order to confer catalytic competency. The archaeal Rpn11 homologue therefore likely represents an interesting intermediate in the evolution of these ubiquitin-linkage modulating systems. Furthermore, it has recently been demonstrated that the zinc-chelating activity of the antibiotic thiolutin inhibits the JAMM metalloprotease activity of the eukaryotic Rpn11 proteasome-complex homologues, the deNEDDylation function of the JAMM metalloproteases Csn5 (of the COP9 signalosome) and also the activities of the eukaryotic BRCC36 enzyme (a deubiquitylase specific to K63 linkages in the BRCA1/BRCA2 complex)[54]. It therefore seems likely that thiolutin will also inhibit the archaeal Rpn11 homologues given the wide-ranging activity of this antibiotic.

The clustering, often in an operonic arrangement, of the proteins involved in the ubiquitylation cascade found in the *Aigarchaeota* and also across the Asgard archaea is indicative that this arrangement was a prokaryotic invention and does not appear to have been acquired via horizontal gene transfer (HGT) from a eukaryotic source[11, 27, 55]. Sporadic distribution of distant E2 and E3 homologues has also been identified in several bacterial species to date[32], including some examples of complete operons[27], indicative of HGT of the ubiquitylation components between archaeal and bacterial species. However, the high degree of sequence and structural identity of the archaeal components to the well-characterised eukaryotic counterparts strongly indicates

that the ubiquitylation system adopted in early eukaryogenesis evolved from a compact operon similar to the system recently identified in *C. subterraneum*[25] (see Supplementary Notes 3 and 7 for further discussion). In the current study, this ubiquitin modifier system has been characterised biochemically, and shown to function in a manner evocative of the eukaryotic ubiquitylation cascade. The emergence of a functional ubiquitylation system has been proposed to have been one of the primary requirements to permit the evolution of cellular complexity and the eventual emergence of the first eukaryotic cell[9]. Clearly, the subsequent expansion of the ubiquitylation system from a compact ancestral arrangement to the current sophisticated pathways operating in complex eukaryotic organisms, utilising hundreds or even thousands of E3 enzymes, was concomitant with the development of advanced cellular mechanisms during the evolution of eukaryotes[11, 27, 55]. Indeed, the identification of various eukaryotic-like signature proteins including the E1/E2/E3 ubiquitylation cascade in the Asgard archaea superphylum[26, 28] appears to provide our first glimpses into the emergence of these vital cellular systems.

## Methods

**Protein expression and purification.** The genes for Candidatus *Caldiarchaeum subterraneum rpn11 (Csub1473)*, ubiquitin (*Csub1474*), E2-like (*Csub1475*), E1-like (*Csub 1476*) and *srfp (E3-like) (Csub1476)* were synthesised using the GeneArt Genestrings service (ThermoFisher) and the ORFs then subsequently amplified by PCR using the primers described in Supplementary Table 3. Both the full-length ubiquitin (including the C-terminal pro-peptide) and a product truncated immediately after the C-terminal di-glycine motif (Ubq-GG) were amplified. The ubiquitin-GFP fusion construct was generated by amplifying Ubq-GG using the CSUB_1474UBIforNdeI and CSUB1474_UBIrevNdeIGG primers and the amplified product was subsequent cloned into the *NdeI* site immediately upstream of a previously generated (sf)GFP (Sandia Biotech) construct cloned into pET30[1]. The *Sulfolobus acidocaldarius mre11 (Saci0052)* and *rad50 (Saci0051)* ORFs were amplified by PCR from *S. acidocaldarius* DSM639 genomic DNA using the primers described in Supplementary Table 3 and cloned into pET28a and site 2 of pDUET (Novagen), respectively. Amplified genes were cloned into the plasmids with the restriction sites placing the ORFs in frame with the plasmid-encoded hexa-histidine tags. The ubiquitin-T46D, E2-PY67/68AA (loop 4), E2-PS103/104AA (loop7), E2-PS103/104AQ (loop7), srfp (E3-like)-W58A, srfp (E3-like)-I30Q and srfp (E3-like)-R69A site-directed mutants were generated using the QuikChange system (Agilent), using the primers indicated in Supplementary Table 3 to modify the native clones. All clones were verified by DNA sequencing of the complete ORF.

Proteins were expressed in Rosetta (DE3) pLysS *Escherichia coli* cells (Novagen). Cultures were grown at 37 °C to an $OD_{600}$ of 0.3 then cooled to 25 °C and further grown to an $OD_{600}$ of 0.6 and induced overnight with 0.33 mM IPTG. Cells expressing the *C. subterraneum* proteins were harvested by centrifugation, resuspended in 20 mM Tris-HCl (pH 8), 300 mM NaCl, 5% glycerol, 0.05% β-mercaptoethanol, while cells expressing the *S. acidocaldarius* Mre11/Rad50 complex were instead resuspended in 100 mM Tris-HCl pH 8, 300 mM NaCl, 10% glycerol. 1X EDTA-free protease inhibitors (Complete cocktail, Roche) were added (with the exception of *C. subterraneum* Rpn11 protease) and cells were lysed by sonication and heat clarified at 60 °C for 20 min before centrifugation at (14,000 r.p.m. for 10 min) to remove insoluble material. Supernatants were filtered and then applied by gravity flow to a column of Ni-NTA agarose (Qiagen). The columns were washed with resuspension buffer and then resuspension buffer plus 15 mM imidazole. Proteins were then eluted in resuspension buffer plus 500 mM imidazole. At this stage the *S. acidocaldarius* Mre11/Rad50 preparations were incubated with 500 units of Benzonase nuclease (Sigma) for one hour. Fractions containing the purified proteins were pooled and concentrated before running a size-exclusion chromatography step over a Superdex 200 16/600 column (GE Healthcare), or an S75 16/600 column (GE Healthcare) in 20 mM Tris-HCl pH 8, 300 mM NaCl, 5% glycerol, 0.5 mM dithiothreitol. Fractions containing the purified proteins were pooled, concentrated, aliquoted and flash frozen in liquid $N_2$. Protein concentrations were quantified by UV spectrophotometry.

Untagged proteins were prepared as above but prior to the final size-exclusion chromatography step the tagged protein was dialysed against thrombin cleavage buffer (20 mM Tris-HCl pH 8, 150 mM NaCl, 2.5 mM $CaCl_2$, 5% glycerol) overnight at room temperature. His-tags were removed by cleavage with thrombin (Novagen) at room temperature for two hours. The cleaved sample was then applied by gravity flow to a column containing Ni-NTA agarose (Qiagen) and washed with 20 mM Tris-HCl pH 8, 300 mM NaCl, 5% glycerol 0.5 mM dithiothreitol plus 15 mM imidazole. The untagged proteins were further purified by size-exclusion chromatography as described above.

**Pro-ubiquitin cleavage with the Rpn11 JAMM metalloprotease.** 50 μg untagged *C. subterraneum* pro-ubiquitin was incubated either with or without 50 μg N-terminally His-tagged Rpn11 in 300 μl reaction buffer (20 mM Tris-HCl pH 8, 150 mM NaCl, 5% glycerol, 5 mM MgCl$_2$, 1 mM dithiothreitol (DTT)) for 15 min at 60 °C. Products were analysed by sodium dodecyl sulphate polyacrylamide gel electrophoresis (SDS-PAGE) and visualised by Coomassie staining. Cleaved and uncleaved products were examined further by tandem mass-spectrometry as described below. For the downstream ubiquitylation assays the His-tagged Rpn11 protein was separated from the cleaved ubiquitin product by Ni-NTA IMAC. 100 μl of nickel-agarose slurry (Qiagen) was washed three times in 500 μl reaction buffer and then applied to the cleaved reaction and incubated at room temperature with agitation for 15 min. The supernatant, containing the cleaved ubiquitin, was then aspirated and used in the subsequent ubiquitylation reactions.

**Ubiquitylation reactions.** For the E1 and E2 autoubiquitylation assays 50 μg of full length pro-ubiquitin (Ubq-FL), pro-ubiquitin cleaved with Rpn11 (Ubq-CLV), or ubiquitin truncated after the C-terminal di-glycine motif (Ubq-GG) were incubated in the presence or absence of 50 μg E1-like protein with or without the addition of 50 μg of the E2-like protein in 200 μl reaction buffer (20 mM Tris-HCl pH 8, 150 mM NaCl, 5% glycerol, 5 mM MgCl$_2$, 1 mM dithiothreitol) for 20 min at 60 °C (or 60 min at 45 °C) in the presence or absence of 3.3 mM ATP. For the full E1/E2/srfp (E3) cascade assays 50 μg Ubq-GG, 10 μg E1-like protein, 10 μg E2-like protein, 50 μg srfp (E3-like) ubiquitylation cascade protein and 30 μg of the *S. solfataricus* Mre11/Rad50 substrate were incubated in 200 μl reaction buffer (20 mM Tris-HCl pH 8, 150 mM NaCl, 5% glycerol, 5 mM MgCl$_2$, 1 mM dithiothreitol) for 60 min at 60 °C in the presence or absence of 2.5 mM ATP (with addition of fresh ATP at 15, 30 and 45 min). For the auto-ubiquitylation assays shown in Fig. 1 only the E1-like enzyme or a combination of the E1-like and E2-like enzymes were added to the reaction. For the ubiquitylation of the *S. solfataricus* Mre11/Rad50 reactions the srfp (E3-like) protein was also added to reconstitute the full ubiquitylation cascade. Negative control reactions omitting one of each of the three ubiquitylation enzymes were prepared alongside the cascade reaction containing all three components. Products were analysed by SDS-PAGE, followed by Coomassie staining or Western analysis, or tandem mass-spectrometry. Primary antibodies (either custom polyclonal rabbit antibody raised against the *C. subterraneum* ubiquitin protein or *S. acidocaldarius* Rad50 protein (Covalab) were used at 1:1000 dilution or a commercial anti-6X-histidine mouse antibody (BD Pharmigen Catalogue No. 552565) was used at 1:200 dilution in 1 X TBST. Horseradish peroxidase (HRP)-conjugated secondary antibodies (either Thermo Fisher Scientific anti-mouse [Cat No. 32430] used at 1:1000 dilution or Thermo Fisher Scientific anti-rabbit [Cat No. A16110] used at 1:10,000 dilution) were incubated at R/T for 1 h in 1 X TBST and bands were detected using an Amersham ECL Western blotting kit.

**Rpn11/JAMM deconjugation of ubiquitin modified lysines.** Following the ubiquitylation of 30 μg *S. solfataricus* Mre11/Rad50 substrate using N-terminally His$_6$-tagged UBQ-GG and the E1-like, E2-like and srfp (E3-like) components (carried out as described above), the ubiquitylated Rad50 products were transferred to an ATP-free reaction buffer (20 mM Tris-HCl pH 8, 150 mM NaCl, 5% glycerol, 5 mM MgCl$_2$, 1 mM DTT) via Ni-NTA IMAC. 250 μl of nickel-agarose slurry (Qiagen) was washed three times in 500 μl reaction buffer and then applied to the ubiquitylation reaction and incubated at room temperature with agitation for 15 min. The beads were then washed once in reaction buffer without ATP plus 15 mM imidazole and twice in 500 μl reaction buffer without ATP or imidazole. Following re-suspension of the Ni-agarose bound products in 300 μl reaction buffer, 50 μg N-terminally His$_6$-tagged Rpn11 was added and the reaction incubated at 60 °C for 15 min. The products were washed three times in 500 μl reaction buffer and the products eluted with 150 μl reaction buffer plus 500 mM imidazole. The products were analysed by SDS-PAGE and visualised by Coomassie staining or alternatively transferred to nitrocellulose membranes by western blot probed with either anti-ubiquitin or anti-histidine antibodies.

**Rpn11/JAMM cleavage of a ubiquitin-(sf)GFP fusion.** 50 μg superfolder-(sf)GFP (Sandia Biotech) or 50 μg 1xUbq-(sf)GFP (an N-terminal fusion of the *C. subterraneum* ubiquitin moiety joined via the C-terminal di-glycine motif to the N-terminus of the (sf)-GFP) was incubated with 50 μg Rpn11 in 150 μl reaction buffer (20 mM Tris-HCl pH 8, 150 mM NaCl, 5% glycerol, 5 mM MgCl$_2$, 1 mM dithiothreitol) for 15 min at 60 °C. The products were analysed by SDS-PAGE and visualised by Coomassie staining.

**Size-exclusion chromatography.** Physical interaction between the E1-like and E2-like proteins was examined by size-exclusion chromatography using an analytical Superdex S200 HR 10/300 column (GE Healthcare). The E1/E2 complex was formed at 60 °C, before the gel filtration analysis by mixing together 250 μg of each protein in a final volume of 500 μl gel filtration buffer (20 mM Tris [pH 8.0], 300 mM NaCl, 5% glycerol, 1 mM DTT). Reactions were subsequently spun at 16,000 *g* in a benchtop centrifuge for 5 min to remove any precipitated material, before loading onto the size exclusion chromatography column. 0.5 ml fractions were collected and resolved by SDS-PAGE, on 15% polyacrylamide gels. The proteins were visualised with Coomassie stain.

**GeLC-mass spectrometry and MS data analyses.** The *C. subterraneum* E1 and E2 bands and the *S. acidocaldarius* Mre11/Rad50 1D gel bands were excised and transferred into a 96-well PCR plate. The gel bands were cut in half and then each half was cut into 1 mm$^2$ pieces. The two halves were then destained and half the gel pieces were reduced (DTT) and alkylated (iodoacetamide) and half were not reduced and alkylated. The reason for not reducing and alkylating the samples was because the procedure may have interfered with the potential TVGG at cysteine residues. The samples were then subjected to enzymatic digestion with trypsin overnight at 37 °C. A similar procedure was followed for the Rpn11-cleaved ubiquitin bands except that both gel pieces were reduced and alkylated and chymotrypsin replaced trypsin. After digestion, the supernatant was pipetted into a sample vial and loaded onto an autosampler for automated LC-MS/MS analysis.

All LC-MS/MS experiments were performed using a nanoAcquity UPLC (Waters Corp., Milford, MA) system and an LTQ Orbitrap Velos hybrid ion trap mass spectrometer (Thermo Scientific, Waltham, MA). Separation of peptides was performed by reverse-phase chromatography using a Waters reverse-phase nano column (BEH C18, 75 mm i.d. × 250 mm, 1.7 mm particle size) at flow rate of 300 nL/min. Peptides were initially loaded onto a pre-column (Waters UPLC Trap Symmetry C18, 180 mm i.d × 20 mm, 5 mm particle size) from the nanoAcquity sample manager with 0.1% formic acid for 5 min at a flow rate of 5 mL/min. After this period, the column valve was switched to allow the elution of peptides from the pre-column onto the analytical column. Solvent A was water + 0.1% formic acid and solvent B was acetonitrile + 0.1% formic acid. The linear gradient employed was 5–40% B in 60 min.

The LC eluant was sprayed into the mass spectrometer by means of a New Objective nanospray source. All *m/z* values of eluting ions were measured in the Orbitrap Velos mass analyser, set at a resolution of 30,000. Data dependent scans (Top 20) were employed to automatically isolate and generate fragment ions by collision-induced dissociation in the linear ion trap, resulting in the generation of MS/MS spectra. Ions with charge states of 2+ and above were selected for fragmentation.

Post-run, the data was processed using Protein Discoverer (version 1.4., ThermoFisher). Briefly, all MS/MS data were converted to mgf files and the files were then submitted to the Mascot search algorithm (Matrix Science, London UK) and searched against a custom database containing the sequences of the *C. subterraneum* E1, E2, Ubq-FL (full length pro-ubiquitin including the CGEPIRRAA propeptide) and UBQ-GG (mature ubiquitin ending with the di-glycine motif) and the *S. acidocaldarius* Mre11/Rad50 proteins along with a number of contaminant sequences such as keratins and the digestion enzymes. The search settings employed were a fixed modification of carbamidomethyl and variable modifications of oxidation (M), deamidation (N/Q) and the ubiquitin modification TVGG (C/K). The peptide and fragment mass tolerances were set to 25 ppm and 0.8 Da, respectively. A significance threshold value of $p < 0.05$ and a peptide cut-off score of 20 were also applied.

**Structural modelling.** Amino-acid sequences of the *C. subterraneum* E1-like, E2-like and srfp (E3-like) ubiquitylation enzymes were submitted to the I-TASSER[35–37] server for folding prediction by homology (see Supplementary Information for additional information and confidence scores). PHYRE2 (Protein Homology/analogY Recognition Engine V 2.0) searches[46] (see Supplementary Information) were also performed to identify eukaryotic structural homologues of the *C. subterraneum* E1-like, E2-like and srfp (E3-like) with 99.8–100% confidence (this is a confidence score that is representative of the probability (ranging from 0 to 100%) that the matching sequences between the input and template is a true homology).

**Identification of gene clusters.** All archaeal assemblies and proteomes were downloaded from NCBI (Feb-2017). For those assemblies lacking of protein annotations, proteins were predicted using prodigal (version 2.6.3, default parameters). A representative subset of proteomes were assigned to existing archaea-specific clusters of orthologous genes (arCOGs) Makarova et al., 2015[56] and new arCOGs were created using proteins without any assignment as described previously in Spang et al. 2015[26]. ArCOGs containing putative E1, E2, E3, Ub and JAB- proteins were selected, and aligned individually using mafft-linsi (version 7.271, default parameters)[57]. Note that this selection is not exhaustive and there could be other proteins belonging to the ubiquitin machinery assigned to different arCOGs, especially in the case of E3. It is also not specific, as some other related proteins families could be included in this orthologous groups (for example, the arCOG containing E1 proteins also include other ThiF proteins). These alignments were used to identify putative-ubiquitin-related proteins from all the proteomes downloaded from NCBI (psiblast -evalue 1e-6, version 2.2.30+). For each taxa, clusters were identified if there were hits to different UB-related proteins (E1, E2, E3, Ub or JAB) in close proximity (less than 5 proteins away). InterPro-domains were assigned to proteins within the selected clusters (InterProScan—pathways—goterms—iprlookup, version 5.22–61.0). C-terminal ubiquitin-like (UBL) domains were also identified by CD-search analysis online through https://www.ncbi.nlm.

nih.gov/Structure/bwrpsb/bwrpsb.cgi with default parameters, or by the PHYRE2 protein fold recognition server (http://www.sbg.bio.ic.ac.uk/phyre2/html/page.cgi?id=index)[46]. Clusters were drawn using genoPlotR in R (version 3.2.2)[58] and prepared in Adobe Illustrator.

**Phylogenetic analysis of E1-like protein homologues**. The sequence selection from Zaremba-Niedzwiedzka et al.[28] (based on Burroughs et al.[38]) was used as a backbone to distinguish between E1-like families. E1-like candidates within the clusters were identified based on the presence of the IPR000594 InterPro domains and added to the backbone. Sequences were aligned using mafft-linsi (--reorder)[57] and a quick phylogeny was inferred using FastTree (-lg)[59]. This phylogeny was used to manually remove repeated sequences and long-branches. The remaining sequences were aligned and trimmed (232 sites) using maft-linsi (--reorder)[57] and trimAl (version 1.4.rev15, --gappyout)[60] respectively. Iqtree[61] was used to test the evolutionary models and reconstruct the phylogeny (version 1.5.3, -m TESTNEW −mset LG −madd LG + C20,LG + C30, LG + C40,LG + C50, LG + C60 −bb 1000). The model selected by iqtree was LG + R8.

**Phylogenetic analysis of E2-like protein homologues**. Proteins containing the InterPro domain IPR000608 were downloaded from UniProt. In order to reduce the amount of data, sequences with more than 90% of identity were removed in the case of bacterial, viral, eukaryotic and metagenomes sequences and, in the case of the eukaryotic sequences, just the ones marked as "reviewed" were considered. Sequences were aligned using mafft-linsi (--reorder)[57] and a quick phylogeny was inferred using FastTree (-lg)[59]. This phylogeny and alignment were used to manually remove repeated sequences, long-branches and more distant homologous. The remaining sequences were aligned using maft-linsi (--reorder)[57]. Ambiguous aligned-sites at the ends were manually removed and the resulting alignment was further trimmed using trimAl (--gappyout)[60] (141 sites). Iqtree[61] was used to test the evolutionary models and reconstruct the phylogeny (-m TESTNEW −mset LG −madd LG + C20,LG + C30, LG + C40,LG + C50, LG + C60 −bb 1000). The model selected by iqtree was LG + R6.

**Data availability**. The data that support the findings of this study are included in this published article (and its Supplementary Information files) or available from the corresponding author upon reasonable request.

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

## Acknowledgements

This work was supported by an Isaac Newton Trust Research Grant (Trinity College and Department of Biochemistry, Cambridge) and start-up funds from the Division of Biomedical and Life Sciences (Lancaster University) to N.P.R. A.C. and H.A. were BIOL387 undergraduate project students from the Division of Biomedical and Life Sciences, Lancaster University (Summer 2017). The work was also supported by grants of the Swedish Research Council (VR grant 621-2009-4813), the European Research Council (ERC Starting grant 310039-PUZZLE_CELL) and the Swedish Foundation for Strategic Research (SSF-FFL5) to T.J.G.E.

## Author contributions

N.P.R. designed the study and drafted the manuscript. R.H.J., J.A.H., M.J.D. and N.P.R. collected and analysed the biochemical and proteomic data. R.H.J. and N.P.R. performed the structural predictions. J.A.H. and M.J.D. collected the mass-spectrometry data sets. A.E. and H.A. assisted during the generation and purification of the mutant ubiquitin, E2-like and srfp (E3-like) proteins. E.F.C. and T.J.G.E. analysed the genomic data and performed phylogenetic analyses. R.H.J., E.F.C., T.J.G.E. and N.P.R. discussed the findings and contributed to writing the manuscript.

## Additional information

**Competing interests:** The authors declare no competing financial interests.

