## [Peer Review File · Nature Communications]

Reviewers' comments:

Reviewer #1 (Remarks to the Author):

In their manuscript James et al reconstitute a eukaryotic-like ubiquitination system from a thermophilic archaeal organism. Their study provides experimental evidence that the previously identified eukaryotic-like E1 and E2 enzymes along with the suggested E3 RING-type ligase called SrfP form a complete ubiquitylation cascade capable of ubiquitylating a heterologous protein complex from another archaeon that had previously been shown to be targeted by sampylation (modification with small archaeal modifier proteins). As a modifier the authors identified the ubiquitin-homolog in this organism in its pro-form along with the enzyme that processes it to reveal the canonical C-terminal di-glycine motif. The authors demonstrate that the cascade works together and all three components are required to bring about ubiquitylation. The study is well-conceived and executed and will provide another piece of the puzzle that is the evolutionary emergence of eukaryotic ubiquitination.

I support publication after minor revision of Figure 1.

Points to address:

- Figure 1b: The shift of pro-Ub to Ub is visible, but nevertheless the bands should be labeled on the right edge of the gel. Also Rpn11 should be indicated on the gel along with the additional two bands that are present. Do the authors know where those bands derive from and what they are?
- Figure 1c and d: Label all protein components on the right edge of the gel (Ub, E1, E2) like it has been done in Figure 2a.
- Figure 1a: The coloring is a bit unfortunate. Even for persons without problems in perceiving red color, the shades of red, orange and pinkish-orange are difficult to distinguish. I suggest revising the coloring scheme.

Reviewer #2 (Remarks to the Author):

Activation and conjugation of Ub like proteins can be traced back to early evolution processes such as bacterial synthesis of vitamin B1 (Thiamine; in humans lack of Thiamine leads to Beriberi or Wernicke-Korsakoff syndrome, which characterized by fluids accumulation, memory loss paralysis, and ultimately death). Similarly, another UbL cascade was shown to be involve in

the synthesis of the molybdenum co-factor in *E. coli* and other prokaryotes (while in these organisms the enzymes that require the cofactor involve in anaerobic respiration, in humans lack of the co-factor lead to defects in neurological development and early childhood death.

It is now clear that the eukaryotic ubiquitylation system arose and diversified from prokaryotic antecedents. However, the link between the two systems is somewhat less described. A putative minimal eukaryotic-like ubiquitylation apparatus encoded within the archaeal genomes opens a window to look on some of these evolutionary steps. In this manuscript Robinson and co-workers identified a nearly full cascade in the thermophilic archaeon *Candidatus 'Caldiarchoaeum subterraneum'* and demonstrated its function *in vitro*. While many questions remained open such as which proteins are targeted by this ubiquitin cascade and how the Ub-signal is decoded into a cellular response within the archaeon, I found this article very interesting and telling sufficient new information for publication in *Nature Communications*. Particularly interesting is the findings on the Rpn11 like DUB that removes the additional residues at the C-ter of Ub such as in high eukaryotes. Recently, Diernfellner and co-workers showed published *Nature Chemical Biology* (May 1st 2017) that the antibiotic thiolutin inhibits Rpn11 deubiquitylation and the COP9 deneddylation activities. I wonder if this commercially available antibiotic also inhibits the archaeon Rpn11 hence may provide an explanation for its broad antibiotic function both in eukaryotes and prokaryotes.

Other major comments:

1. A key question is if the identified Ub-Like system in *C. subterraneum* is most closely to Ub or maybe to other UbL? Alignment of the UbL sequence with other human, yeast, or other organism models (one can use BLAST against "Model Organisms" UbLs would be helpful to highlight the link between the archaeal UbL system to the eukaryotic one (it seems that indeed the most closed UbL is ubiquitin, but this should be shown).

2. Figure 1:

Fig 1a Some of the colors in are too similar specifically it is very difficult to distinguish between "Ubiquitin-domain protein - IPR029071, IPR000626"

and "Putative E2-like ubiquitinating conjugating enzyme - IPR000608" I am concern that it will be also difficult to distinguish with some other colors.

Fig. 1c lane 5 (at the bottom) it seems that mistakenly the others took an uncleaved (Ubq-FL) instead of cleaved Ub to the assay. Unfortunately, the authors must correct this panel with new gel.

3. The authors should indicate in the if shown SDS-PAGE or WB (this is apply to all figures).

4. Figure 2:

Fig. 2a and 2b move the "I-TASSER" wording to the legend just write "model"

Fig. 2c * The font is too small.

* Some of the lines have no names this is unacceptable.

* Some lines have the same text for example '6E polyubi associated' so what is the differences

between them.

* How come ThiS/ThiF are shown together ThiF is the E1 where is ThiS is the UbL?

Fig. 2d: * Many E2s form dimers. How do the authors know that you do not look on E2 dimerization and co-migration with monomeric E1? SEC with only E2 must be shown to exclude this possibility.

*A calibration of the column will be very helpful.

5. Fig. 3a Residues should be colored in CPK format to better show the atom properties of the Zn⁺² coordination.

6. Figure 4

* Fig 4b should shows structural based sequence alignment with human Ub.

* Fig 4c should show also superposition of the bovine and the *C. subterraneum* Ub structures. The real conserved lysine is actually K29 of the mammalian Ub with K31 of the *C. subterraneum*.

Although they present different rotamers the CA-CA and CB-CB distances are less than 1Å mammalian Ub K63 is not closed to the *C. subterraneum* K66 (NZ-NZ distance = 16.6Å). Actually it is closer to the mammalian K6 (NZ-NZ distance = 4.1Å) but this is probably only by chance and not due to the evolution of Ub.

Similarly, *subterraneum*-K53 is not closed to mammalian-K48 (NZ-NZ distance = 13.9) but closer to mammalian-K27 (6.1Å), but again I do not think this pair is evolutionary conserved. It would be nice to show that the I44 patch is also conserved. Interestingly it seems that the patch is even larger than the one in eukaryotes. I would speculate in the discussion that a Ub-receptor that read the Ub code is hidden in this organism

Minor comments

. Abstract – lane 28 as the target was not identified and consequently our knowledge on the ubiquitylation cascade is in complete, I suggest to replace the ‘sequential ubiquitylation cascade’ with ‘sequential enzymatic cascade’

. Page 2 lane 5 ‘srfp(E3-like)’ space is missing.

. Page 3 lane 30 replace ‘encouraging’ with ‘inducing’

. Page 4 lane 30 change to: ‘...functions in membrane protein trafficking, immune response and DNA repair.’ (and relevant references).

. Methods:

. In the ‘ubiquitylation reactions’ The microgram amounts of the components must be reported.

.

Similarly in the ‘Deconjugation of ubiquitin modified lysine residues by the Rpn11 JAMM metalloprotease’ and in the ‘Cleavage of a ubiquitin-(sf)GFP fusion by the Rpn11 JAMM

metalloprotease'

. The microgram amounts of the components must be reported.

. Structural modelling, the authors claim that their models present 99.8-100% confidence. What does it mean? The authors have to explain what is the meaning of this confidence or to rewrite the sentences that describing this confidence (also in the main text). and in the figure legends (example Fig. 2)

Reviewer #3 (Remarks to the Author):

This study provides experimental validation that the components of a previously predicted eukaryote-like ubiquitin conjugation system from the archaeum *C. subterraneum* can function as a bona fide ubiquitination cascade. To this end the authors recombinantly expressed and purified the archaeal ubiquitin, E1, E2, E3, and DUB variants and demonstrate that the enzymes are active in an in vitro setting. These experimental data are technically sound; the results are novel and important.

An additional claim of the study is that the archaeal enzymes utilize the same key residues and structural features as the eukaryotic ones. In essence, this claim has been made before (Nunoura et al., *Nucleic Acids Res*, 2011). Unfortunately, the study does not provide any experimental data to support this claim, but instead uses sequence alignments and homology modeling - thus considerations that have, at least in part, also been reported before (Nunoura et al., *Nucleic Acids Res*, 2011). In my eyes, this is where the study falls short of its claims and misses an opportunity to raise the impact of the story to a higher level, as would be desirable to warrant publication in *Nature Communications*.

In consequence, I consider the current scope of the study to be somewhat limited and would recommend either submission to a more specialist journal or a substantial experimental enrichment of the study to provide validation of the structural claims.

Major points:

The authors use the I-Tasser server to generate homology models of the archaeal ubiquitination enzymes, which are included as main figures. However, the authors do not utilize these models to reach significant conclusions that could not be drawn from sequence alignments or, presumably, from domain prediction servers.

It appears that the generated homology models serve the major purpose of showing that the archaeal enzymes are structurally highly homologous to the eukaryotic ones. This should be validated experimentally. For instance, the authors speculate that the archaeal RING domain promotes a closed ubiquitin-E2 conformation, as seen for eukaryotic RING E3 enzymes (Page 3, bottom). However, no evidence is provided in support of this claim. To do so, the authors should show that critical residues in the respective binding sites are conserved. It would also be straightforward and required to support this claim experimentally by site-directed mutagenesis and activity assays.

Alternatively, the homology models may be used to identify interesting differences between the archaeal and eukaryotic proteins. If such differences exist, they should be explored experimentally, which would enrich the study significantly.

The second part of the discussion section (Page 6, lines 7-25), though it surely is interesting, reads like a summary of previous publications and does not really contain any new conclusions drawn from the data presented in this study. This may be indicative of the notion that the study may require more experimental substance, as indicated above.

Minor points:

The gel in Figure 2A appears redundant with Figure 1C.

It would be helpful to label all depicted side chains in the structural models.

Fig 3A: It would be clearer to display side chains (rather than side chain and main chain atoms) of the Zn-coordinating residues in the structural models or, alternatively, to color the residues by atom type. It would also be helpful to label the residues according to the schematic on the left.

Figure 3B would be more meaningful if the interaction schematic was shown side by side with the analogous higher eukaryotic one, based on which the indicated contacts were predicted.

The side chains in the I-Tasser model of SRFP that are predicted to coordinate Zn appear misaligned (Figure 3A), which is not unexpected, given that this is a homology model with no Zn present. However, the authors should comment on this, since they make the claim that the critical Zn-coordination sites looks similar to the human RING domain.

With reference to the “major structural features and key catalytic residues that are characteristic of the ... E2 ubiquitin-conjugating (UBC) domains” (page 2/3) the authors cite two rather old

reviews on E2 enzymes, that can not reflect today's structural knowledge of this enzyme family. I recommend that the authors cite an up-to-date review in addition, for instance Stewart et al., Cell Res. 2016. Likewise, I recommend that the authors include a reference to a modern review on the structural mechanisms of E1 enzymes in addition to the primary citation of Olsen et al..

Page 3: "it has been established that eukaryotic RING-domain proteins fold with a cross-brace...forming and interface for E2 binding" - it would be appropriate to cite Zheng et al., Cell 2000 at this point.

When discussing the binding of ubiquitin to the backside of the E2 (Supplementary Material, references 15-18) it would be appropriate to also cite the study by Buetow et al., Mol Cell, 2015.

We would like to thank all three reviewers for their very helpful comments all of which have been addressed in the revised manuscript

Reviewer #1

The study is well-conceived and executed and will provide another piece of the puzzle that is the evolutionary emergence of eukaryotic ubiquitination. I support publication after minor revision of Figure 1.

We are delighted that Reviewer 1 supports publication of our work in *Nature Communications* and have revised Figure 1 following their suggestions (now Revised Figure 2).

Reviewer #2

*I found this article very interesting and telling sufficient new information for publication in *Nature Communications*.*

We are also extremely pleased that Reviewer 2 believes that our work is ideally suited for publication in *Nature Communications*. We have addressed their helpful comments point-by-point below.

*1. A key question is if the identified Ub-Like system in *C. subterraneum* is most closely to Ub or maybe to other UbL? Alignment of the UbL sequence with other human, yeast, or other organism models (one can use BLAST against "Model Organisms" UbLs would be helpful to highlight the link between the archaeal UbL system to the eukaryotic one (it seems that indeed the most closed UbL is ubiquitin, but this should be shown).*

We have now included an alignment of archaeal and eukaryotic ubiquitin homologues in Figure 7e.

2. Figure 1a Some of the colors in are too similar specifically it is very difficult to distinguish between "Ubiquitin-domain protein - IPR029071, IPR000626" and "Putative E2-like ubiquitinating conjugating enzyme - IPR000608" I am concerned that it will be also difficult to distinguish with some other colors. Fig. 1c lane 5 (at the bottom) it seems that mistakenly the others took an uncleaved (Ubq-FL) instead of cleaved Ub to the assay. Unfortunately, the authors must correct this panel with new gel.

Following the comments by both Reviewers 1 and 2 we have now changed the colour scheme of Figure 1 and chosen a color palette designed to provide bold contrast even for readers with colour perception difficulties.

We have also repeated the Rpn11 cleavage of the full-length pro-ubiquitin and the subsequent ubiquitylation assay, as requested by Reviewer 2, and provided a new panel, Figure 2b.

3. The authors should indicate in the if shown SDS-PAGE or WB (this is apply to all figures).

All Coomassie gels and Western blots are now labeled next to each panel in addition to the description already provided in the Figure legend.

4. Fig. 2a and 2b move the "I-TASSER" wording to the legend just write "model".

We have made these changes.

Fig.2c

* The font is too small.

The font size is now increased (revised Figure 4a).

* Some of the lines have no names this is unacceptable.

We have now added names to all lines in the revised Figure 4a.

* Some lines have the same text for example '6E polyubi associated' so what is the differences between them.

Burroughs et al divide them in to two families: 6E.1 and 6E.2 -polyubi associating family. This information has now been added to the figure legend of revised Figure 4a.

* How come ThiS/ThiF are shown together ThiF is the E1 where is ThiS is the Ubl?

ThiS+ThiF homologues refer to ThiF proteins that are clustered with or in an operon with a ThiS homologue (as defined by Burroughs 2008).

Fig. 2d: * Many E2s form dimers. How do the authors know that you do not look on E2 dimerization and co-migration with monomeric E1? SEC with only E2 must be shown to exclude this possibility.

*A calibration of the column will be very helpful.

We have now repeated the interaction study by size exclusion chromatography over an S200 10-300 superdex column including the E2 only control to demonstrate the shift in the profile is due to interaction with the E1 protein and not simply dimerization. Size standards for the column are also given in the revised Figure 4.

5. Fig. 3a Residues should be colored in CPK format to better show the atom properties of the Zn+2 coordination.

Following the suggestions by both Reviewers 2 and 3 we have modified Figure 3A (now revised Figure 5A) to display only the side chains of the

zinc-coordinating residues and coloured the residues by atom-type (CPK format). We have also labeled the residues in accordance with the schematic on the left. In addition we have included an amino-acid alignment of the *C. subterraneum* srfp (E3-like) protein and the eukaryotic E3-RING-like homologues (revised Figure 5c).

6.Figure4

* Fig 4b should shows structural based sequence alignment with human Ub.

A sequence alignment of the archaeal ubiquitins and eukaryotic ubiquitin homologues (including human Ub) is now presented in revised Figure 7e.

* Fig 4c should show also superposition of the bovine and the *C. subterraneum* Ub structures.

A structural superposition of the *C. subterraneum* ubiquitin and eukaryotic human ubiquitin (we assume the reviewer means human rather than bovine Ubq – the sequences are almost identical anyway) is now presented in revised Figure 7d.

The real conserved lysine is actually K29 of the mammalian Ub with K31 of the *C. subterraneum*.

We originally focused on just the three *C. subterraneum* Ub lysine residues that were identified as modified by mass-spectrometry. We have now also highlighted the potential conservation of the mammalian Ub K29 with K31 of the *C. subterraneum*, as noted by the Reviewer, in the revised Figure 7d.

Although they present different rotamers the CA-CA and CB-CB distances are less than 1Å mammalian Ub K63 is not closed to the *C. subterraneum* K66 (NZ-NZ distance = 16.6Å). Actually it is closer to the mammalian K6 (NZ-NZ distance = 4.1Å) but this is probably only by chance and not due to the evolution of Ub.

We have made a slight modification to the main text accordingly.

Similarly, subterraneum-K53 is not closed to mammalian-K48 (NZ-NZ distance = 13.9) but closer to mammalian-K27 (6.1Å), but again I do not think this pair is evolutionary conserved.

We have made a slight modification to the main text accordingly.

It would be nice to show that the I44 patch is also conserved. Interestingly it seems that the patch is even larger than the one in eukaryotes. I would speculate in the discussion that a Ub-receptor that read the Ub code is hidden in this organism

The hydrophobic patch (centred on I44 in the human protein) is now presented in Figure 7f alongside the equivalent hydrophobic patch in the *C. subterraneum* protein..

Minor comments

Abstract – lane 28 as the target was not identified and consequently our knowledge on the ubiquitylation cascade is incomplete, I suggest to replace the ‘sequential ubiquitylation cascade’ with ‘sequential enzymatic cascade’

We will let the editor decide which expression should be used in the final publication.

Page 2 lane 5 ‘srfp(E3-like)’ space is missing.

Amended (and two other instances).

Page 3 lane 30 replace ‘encouraging’ with ‘inducing’

Replaced as suggested.

Page 4 lane 30 change to: ‘...functions in membrane protein trafficking, immune response and DNA repair.’ (and relevant references).

The suggested change has been made.

.Methods:

. In the ‘ubiquitylation reactions’ The microgram amounts of the components must be reported.

These microgram amounts have now been reported more clearly in the main methods section.

. Similarly in the ‘Deconjugation of ubiquitin modified lysine residues by the Rpn11 JAMM metalloprotease’ and in the ‘Cleavage of a ubiquitin-(sf)GFP fusion by the Rpn11 JAMM metalloprotease’

. The microgram amounts of the components must be reported.

These microgram amounts were already described in the main methods section. We have slightly modified this section to ensure that this is more clearly presented.

. Structural modelling, the authors claim that their models present 99.8-100% confidence. What does it mean? The authors have to explain what is the meaning of this confidence or to rewrite the sentences that describing this confidence (also in the main text) and in the figure legends (example Fig. 2)

These PHYRE2 confidence limits are now described further in the main methods section and Supplementary Information.

Additional:

Recently, Diernfellner and co-workers showed published Nature Chemical Biology (May 1st 2017) that the antibiotic thiolutin inhibits Rpn11 deubiquitylation and the COP9 deneddylation activities. I wonder if this commercially available antibiotic also inhibits the archaeon Rpn11 hence may provide an explanation for its broad antibiotic function both in eukaryotes and prokaryotes.

This is an interesting point and will certainly be worthwhile to investigate this further in future studies. We have mentioned this point and cited the relevant study in the revised main text.

Reviewer #3

These experimental data are technically sound; the results are novel and important.

We are pleased that Reviewer 3 recognizes that our findings are novel and important, concurring with the views of Reviewers 1 and 2.

An additional claim of the study is that the archaeal enzymes utilize the same key residues and structural features as the eukaryotic ones. In essence, this claim has been made before (Nunoura et al., Nucleic Acids Res, 2011). Unfortunately, the study does not provide any experimental data to support this claim, but instead uses sequence alignments and homology modeling - thus considerations that have, at least in part, also been reported before (Nunoura et al., Nucleic Acids Res, 2011).

The authors use the I-Tasser server to generate homology models of the archaeal ubiquitination enzymes, which are included as main figures. However, the authors do not utilize these models to reach significant conclusions that could not be drawn from sequence alignments or, presumably, from domain prediction servers.

It appears that the generated homology models serve the major purpose of showing that the archaeal enzymes are structurally highly homologous to the eukaryotic ones. This should be validated experimentally. For instance, the authors speculate that the archaeal RING domain promotes a closed ubiquitin-E2 conformation, as seen for eukaryotic RING E3 enzymes (Page 3, bottom). However, no evidence is provided in support of this claim. To do so, the authors should show that critical residues in the respective binding sites are conserved. It would also be straightforward and required to support this claim experimentally by site-directed mutagenesis and activity assays.

While the study by Nunoura *et al.* does indeed provide the first report of the operonic arrangement of the archaeal ubiquitin, E1-like, E2-like and srfp components and includes some amino-acid sequence alignments with eukaryotic homologues, there is no discussion of the key conserved residues or the mechanism by which the ubiquitylation cascade might

operate. Indeed, this study mainly reports on the first metagenomic assembly of *C. subterraneum* and does not describe the cascade brought about by these ubiquitylation components in significant depth.

Our study demonstrates the first biochemical purification of these components and the subsequent reconstitution of this pathway, and we demonstrate that the ubiquitylation process requires all three of the E1-like, E2-like and srfp proteins. Our structural modeling (which has not been published elsewhere to date) helps to illustrate which of the key conserved residues are predicted to be critical to the process.

We should also point out that the figure of the *C. subterraneum* ubiquitin moiety (now with new views in the modified Figure 7) is not a model but a deposited NMR structure and this has not been published elsewhere to date.

Therefore, in addition to the biochemical reconstitution of the archaeal ubiquitylation cascade, our work also provides the first comprehensive prediction of the likely structural and functional mechanism of the archaeal ubiquitin modification process, although clearly there is now considerable scope for future structural studies utilising this system.

Following the reviewer's practicable request for experimental verification of our predictions we have now generated seven site-directed mutants, and subsequently expressed and purified these proteins and tested them in ubiquitylation cascade assays. The resultant data are supportive of the predictions inferred from our modeling. The outcome of these substantive new analyses are presented in revised Figure 6c. The mutants investigated are E2-PY6768AA, E2-PS103104AA, E2-PS103104AQ, ubiquitin-T46D, srfp-W58A, srf-I30Q, srfp-R69A and all results are discussed fully in the revised manuscript.

Minor points:

The gel in Figure 2A appears redundant with Figure 1C.

We have removed this gel from Figure 2A following the Reviewer's suggestion

It would be helpful to label all depicted side chains in the structural models.

We have now labeled all of the depicted side chains in the revised structural Figures.

Fig 3A: It would be clearer to display side chains (rather than side chain and main chain atoms) of the Zn-coordinating residues in the structural models or

alternatively, to color the residues by atom type. It would also be helpful to label the residues according to the schematic on the left.

Following the suggestions by both Reviewers 2 and 3 we have modified Figure 3a (now revised Figure 5a) to display only the side chains of the zinc-coordinating residues and coloured the residues by atom-type (CPK format). We have also labeled the residues in accordance with the schematic on the left. In addition we have included an amino-acid alignment of the *C. subterraneum* srfp protein with eukaryotic E3-RING-like homologues.

Figure 3B would be more meaningful if the interaction schematic was shown side by side with the analogous higher eukaryotic one, based on which the indicated contacts were predicted.

We have included an additional panel (revised Figure 5c), which shows amino-acid alignments of the loop 4 and loop 7 regions of the E2-enzymes and the RING-domain of the E3-like proteins of the *C. subterraneum* proteins eukaryotic homologues. These motifs are clearly conserved in both the archaeal and eukaryotic systems.

The side chains in the I-Tasser model of SRFp that are predicted to coordinate Zn appear misaligned (Figure 3A), which is not unexpected, given that this is a homology model with no Zn present. However, the authors should comment on this, since they make the claim that the critical Zn-coordination sites looks similar to the human RING domain.

This was previously noted in Supplementary Figure 4. We have now also commented on this in the main text and included E3-like protein amino acid sequence alignments as part of the revised Figure 5c.

With reference to the “major structural features and key catalytic residues that are characteristic of the ... E2 ubiquitin-conjugating (UBC) domains” (page 2/3) the authors cite two rather old reviews on E2 enzymes, that can not reflect today’s structural knowledge of this enzyme family. I recommend that the authors cite an up-to-date review in addition, for instance Stewart et al., Cell Res. 2016. Likewise, I recommend that the authors include a reference to a modern review on the structural mechanisms of E1 enzymes in addition to the primary citation of Olsen et al..

The Stewart reference was originally cited in the Supplementary Materials but we have now also made reference to this review in the main text. In addition to the citation of Olsen et al to describe the E1 enzyme activities, we have also cited Cappadocia and Lima (2017) following the Reviewer’s suggestion.

Page 3: “it has been established that eukaryotic RING-domain proteins fold with a

cross-brace...forming and interface for E2 binding" - it would be appropriate to cite Zheng et al., Cell 2000 at this point.

We have now cited this work at the relevant point in the main text.

When discussing the binding of ubiquitin to the backside of the E2 (Supplementary Material, references 15-18) it would be appropriate to also cite the study by Buetow et al., Mol Cell, 2015.

The additional citation has now been referenced at the relevant points in the Supplementary Materials.

Reviewers' Comments:

Reviewer #1 (Remarks to the Author):

With their revised document the authors have addressed my comments. The figures have gained in clarity and also the text better incorporates the supplementary data. I would also like to point out that the experimental mutant data verifying some of the predictions have greatly improved the manuscript.

I find the manuscript in its current form suitable for publication.

Reviewer #2 (Remarks to the Author):

The authors perfectly addressed the requested points and therefore I find the manuscript now ready for publication in Nature communications.

Also, I found out that I made a mistake in my comment regarding K53 as I confused it with K68 (probably due to the differences in the actual numbers in the PDB (K56 and K76 respectively)).

Reviewer #3 (Remarks to the Author):

In the revised version of the manuscript the authors have addressed most of my concerns and included additional mutational analyses to validate their claims. I, therefore, recommend that the manuscript is accepted for publication.